# Paracrine cellular senescence exacerbates biliary injury and impairs regeneration

Sofia Ferreira-Gonzalez[1], Wei-Yu Lu[1], Alexander Raven[1], Benjamin Dwyer[1], Tak Yung Man[1], Eoghan O'Duibhir[1], Philip J. Starkey Lewis[1], Lara Campana [1], Tim J. Kendall [2], Thomas G. Bird[1,2,3], Nuria Tarrats[4], Juan-Carlos Acosta [4], Luke Boulter [5] & Stuart J. Forbes[1]

Cellular senescence is a mechanism that provides an irreversible barrier to cell cycle progression to prevent undesired proliferation. However, under pathological circumstances, senescence can adversely affect organ function, viability and regeneration. We have developed a mouse model of biliary senescence, based on the conditional deletion of Mdm2 in bile ducts under the control of the Krt19 promoter, that exhibits features of biliary disease. Here we report that senescent cholangiocytes induce profound alterations in the cellular and signalling microenvironment, with recruitment of myofibroblasts and macrophages causing collagen deposition, TGFβ production and induction of senescence in surrounding cholangiocytes and hepatocytes. Finally, we study how inhibition of TGFβ-signalling disrupts the transmission of senescence and restores liver function. We identify cellular senescence as a detrimental mechanism in the development of biliary injury. Our results identify TGFβ as a potential therapeutic target to limit senescence-dependent aggravation in human cholangiopathies.

[1] MRC Centre for Regenerative Medicine, University of Edinburgh, 5 Little France Drive, Edinburgh EH16 4UU, UK. [2] MRC Centre for Inflammation Research, University of Edinburgh, Edinburgh EH16 4TJ, UK. [3] Cancer Research UK Beatson Institute, Glasgow G61 1BD, UK. [4] Edinburgh Cancer Research UK Centre, The Institute of Genetics and Molecular Medicine, University of Edinburgh, Edinburgh EH4 2XR, UK. [5] MRC Human Genetics Unit, Institute of Genetics and Molecular Medicine, University of Edinburgh, Edinburgh EH4 2XU, UK. Correspondence and requests for materials should be addressed to S.J.F. (email: stuart.forbes@ed.ac.uk)

Primary sclerosing cholangitis (PSC) and primary biliary cholangitis (PBC) are the most prevalent type of cholangiopathies, a diverse group of genetic and acquired disorders that affect the biliary population of the liver[1–3]. PSC/PBC have variable prognoses but frequently evolve into end-stage liver disease, with limited treatment options.

The aetiologies remain unclear, although a role of cellular senescence in the development of PSC/PBC has been suggested[4–8]. Senescence is an irreversible cell cycle arrest, driven by dominant cell-cycle inhibitors and characterized by changes in morphology, increased lysosomal content[9], expression of DNA damage response (DDR) factors[10], and the activation of the senescence-associated secretory phenotype (SASP)[11,12].

The SASP is a pro-inflammatory response that activates and reinforces the senescent phenotype in the surrounding cells[12], modulates fibrosis[13] and promotes regeneration[11,14]. The SASP is composed by a variable set of secreted cytokines and chemokines, responsible for the beneficial and deleterious effects of senescence within the tissue[11–18]. TGFβ, a cytokine involved in proliferation, migration, cellular differentiation and cell cycle regulation is a critical element for the establishment of paracrine senescence through the SASP. TGFβ induces and maintains paracrine senescence through a mechanism that generates reactive oxygen species and DDR[12,19] and by inducing p21 expression in a p53 independent manner through the regulation of p27 and p15[11,19–21].

However, despite a number of studies suggesting a potential link between senescence and biliary disease[4–8], it has not been shown whether senescence is actually a driver of the damage rather than solely a consequence[8]. We have therefore investigated the relationship between senescence and biliary disease, focusing on SASP-related mechanisms to explain part of the pathophysiology of PSC/PBC.

Here, we present a model of biliary disease, based on the conditional deletion of Mdm2 in bile ducts under the control of the Krt19 cholangiocyte promoter. In this model, senescent cholangiocytes induce profound alterations in the cellular and signalling microenvironment, with recruitment of myofibroblasts and macrophages causing collagen deposition, TGFβ production and induction of senescence in surrounding cholangiocytes and hepatocytes. Finally, we study how inhibition of TGFβ-signalling disrupts the transmission of senescence and restores liver function. In doing so we provide potential opportunities for therapies based on the disruption of the TGFβ-dependent SASP response.

## Results

**Cellular senescence is an intrinsic feature of PSC/PBC.** Assessment of PSC/PBC human biopsies corroborate the presence of previously described senescence-factors like p21[4] (Fig. 1a). p21 is also expressed in some periportal hepatocytes (Fig. 1a).

Other senescence markers such as p16[8] (Fig. 1b), DCR2[11] (Fig. 1c), γH2A.X[8] (Fig. 1d) and p27[11] (Fig. 1e) are also present in Keratin 19 (K19)-expressing cholangiocytes. Again, some periportal hepatocytes express markers of senescence such as γH2A.X and DCR2 (Supplementary Fig. 1a, b).

PSC and PBC are clinically characterized by increased ductular response (Supplementary Fig. 1c), apoptotic bile ducts (Supplementary Fig. 1d)[4], extensive fibrotic areas (Supplementary Fig. 1e) and copper deposition (Supplementary Fig. 1f), all important factors for the evolution of the disease[1–3].

**Deletion of Mdm2 results in p21-driven senescence.** To test whether senescence could be a pathological driver of such cholangiopathies, we developed a model in which biliary senescence is triggered in cholangiocytes using a tamoxifen-inducible K19-Mdm2$^{flox/flox}$tdTom$^{LSL}$ mice (Fig. 2a).

Murine Double Minute 2 (Mdm2) is a key negative regulator of p53, that positively regulates p21 expression. Therefore, cholangiocyte-specific (via Keratin19CreER$^{T}$) deletion of Mdm2 results in p21 expression in bile ducts (Fig. 2b and Supplementary Fig. 2 a, b).

Low occurrence of proliferating p53-positive cholangiocytes (0.1% Mean ± SEM) (Fig. 2c and Supplementary Fig. 2c) suggests that activation of the p53-p21 axis initiates cell cycle arrest. Expression of other senescence markers like 53BP1, γH2A.X and DCR2 (Fig. 2d) suggests the occurrence of cellular senescence in cholangiocytes. Furthermore, some hepatocytes also express 53BP1, γH2A.X and DCR2 (Fig. 2d) while maintaining Mdm2 expression (Supplementary Fig. 2d). This suggests that the presence of senescent hepatocytes is a consequence of paracrine activity from the cholangiocytes and not due to Cre-recombination or spontaneous loss of Mdm2 in hepatocytes.

Cre activity in senescent cholangiocytes can be traced by the expression of tdTomato (tdTom) (Fig. 2e), which is restricted to cholangiocytes indicating that Cre activity is cholangiocyte specific. The percentage of recombined tdTom-positive cells that proliferate is very low and does not significantly vary over the course of 90 days, suggesting that the cell cycle arrest remains unaltered (Supplementary Fig. 2e). Furthermore, tdTom expression colocalizes with p53 in cholangiocytes (Fig. 2f), suggesting that p53 expression is directly associated with tdTom presence in cholangiocytes.

The percentage of tdTom-positive cells remain unchanged in the liver 90 days post-induction (Supplementary Fig. 3a), accompanied by ductular reaction (Supplementary Fig. 3b, c). Over the course of 90 days, we observed liver disease progression as evidenced by changes in liver biochemistry, including alanine aminotransferase, aspartate aminotransferase and bilirubin (Supplementary Fig. 3d). We did not observe ductopenia (Supplementary Fig. 3e).

Our model also displays p16 in cholangiocytes and some hepatocytes (Supplementary Fig. 4a), suggesting that senescence expands from cholangiocytes towards the parenchyma through paracrine signalling.

Furthermore, K19-Mdm2$^{flox/flox}$tdTom$^{LSL}$ mice display large mononuclear infiltrates (Supplementary Fig. 4b) which were identified as F4/80-positive macrophages (Supplementary Fig. 4c), suggesting the presence of cholangiocyte-dependent paracrine signalling. Twenty-one days after induction, apoptosis occurs in the bile ducts (Supplementary Fig. 4d), which was previously described in human PSC/PBC[4] and could potentially amplify the effect of senescence to exacerbate biliary injury.

We found increased intensity of αSMA-positive cells surrounding the senescent cholangiocytes (Fig. 2g, h and Supplementary Fig. 5a, b, c). We also found increased collagen deposition (Fig. 2i) that progressively accumulates over the course of 90 days (Supplementary Fig. 5d).

As our model induces cellular senescence in the K19-positive epithelia, we examined other organs (kidney and gut) for markers of cellular senescence such as p16 (Supplementary Fig. 6a). Both kidney and gut presented increased levels of fibrosis over the course of 90 days (Supplementary Fig. 6b). We did not find evidence of the involvement of mesothelial cells in our model (Supplementary Fig. 6c).

We also compared the senescence response in Mdm2$^{flox/flox}$ models with different promoters (see Supplementary Fig. 7a for K19-Mdm2$^{flox/flox}$tdTom$^{LSL}$ model, and Supplementary Fig. 7b for OPN-Mdm2$^{flox/flox}$YFP, respectively), both targeting cholangiocytes[22,23]. Both models present similar levels of recombination (Supplementary Fig. 7c) and induction of senescence in

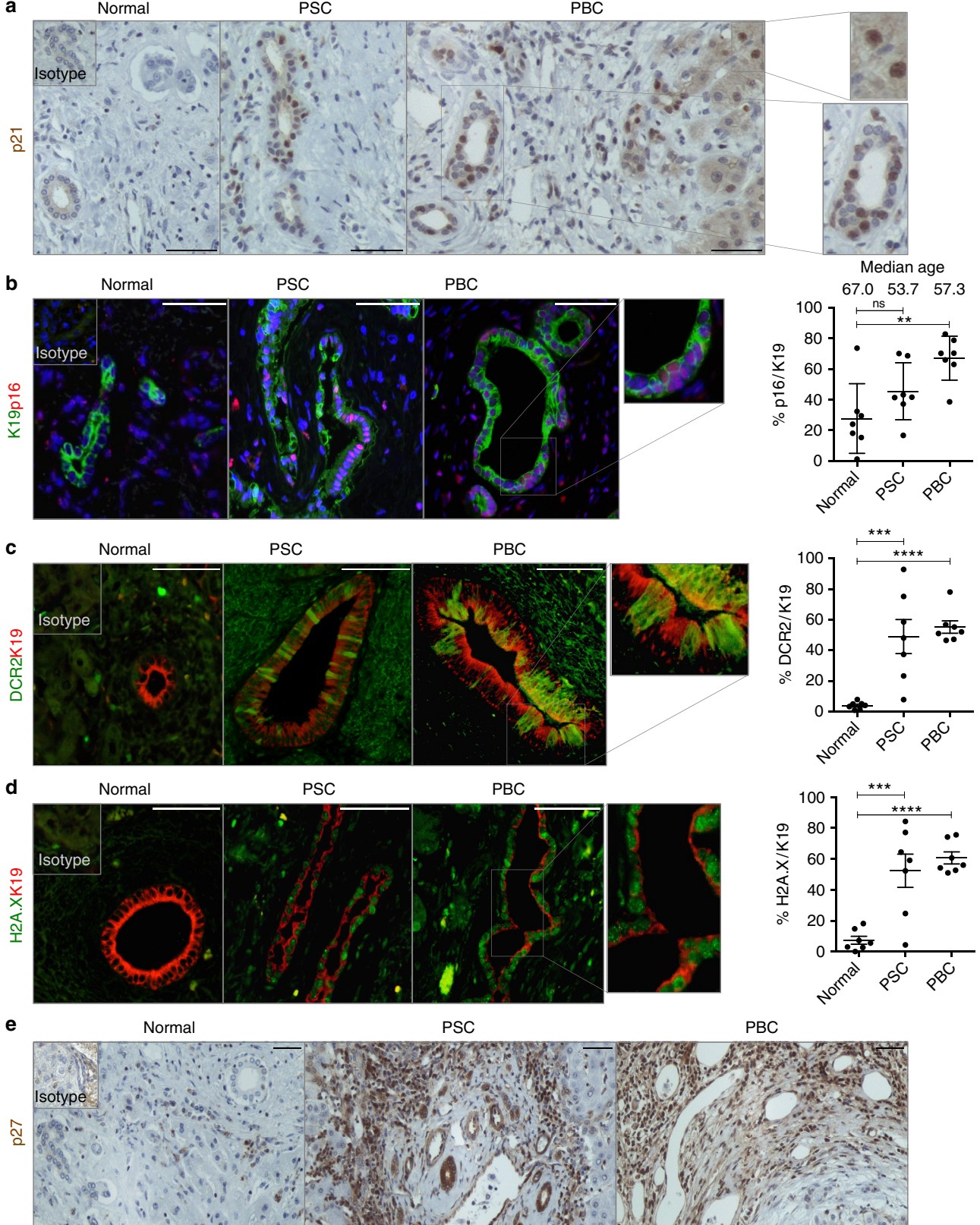

**Fig. 1** Cellular senescence is an intrinsic characteristic of PBC/PSC. Explanted human livers diagnosed as Normal, PSC and PBC respectively (series of $N = 7$ per group). **a** Immunohistochemical detection of p21 in bile ducts. Far right, digital magnification of p21-positive hepatocytes (top) and cholangiocytes (bottom). **b** PSC and PBC show p16 expression (red) in cholangiocytes (green). Far right, quantification of p16 in K19-positive cholangiocytes. Median age of each of the groups is also included in this figure. **c** PSC and PBC show DCR2 expression (green) in cholangiocytes (red). Far right, quantification. **d** PSC and PBC show γH2A. X expression (green) in cholangiocytes (red). Far right, quantification. **e** Immunohistochemical detection of p27 increases in PSC and PBC compared to Normal Liver. ** denotes $p < 0.01$, *** denotes $p < 0.001$, **** denotes $p < 0.0001$ (Mean ± SEM). ANOVA, Sidak's multiple comparisons test. Scale bars = 50 μm

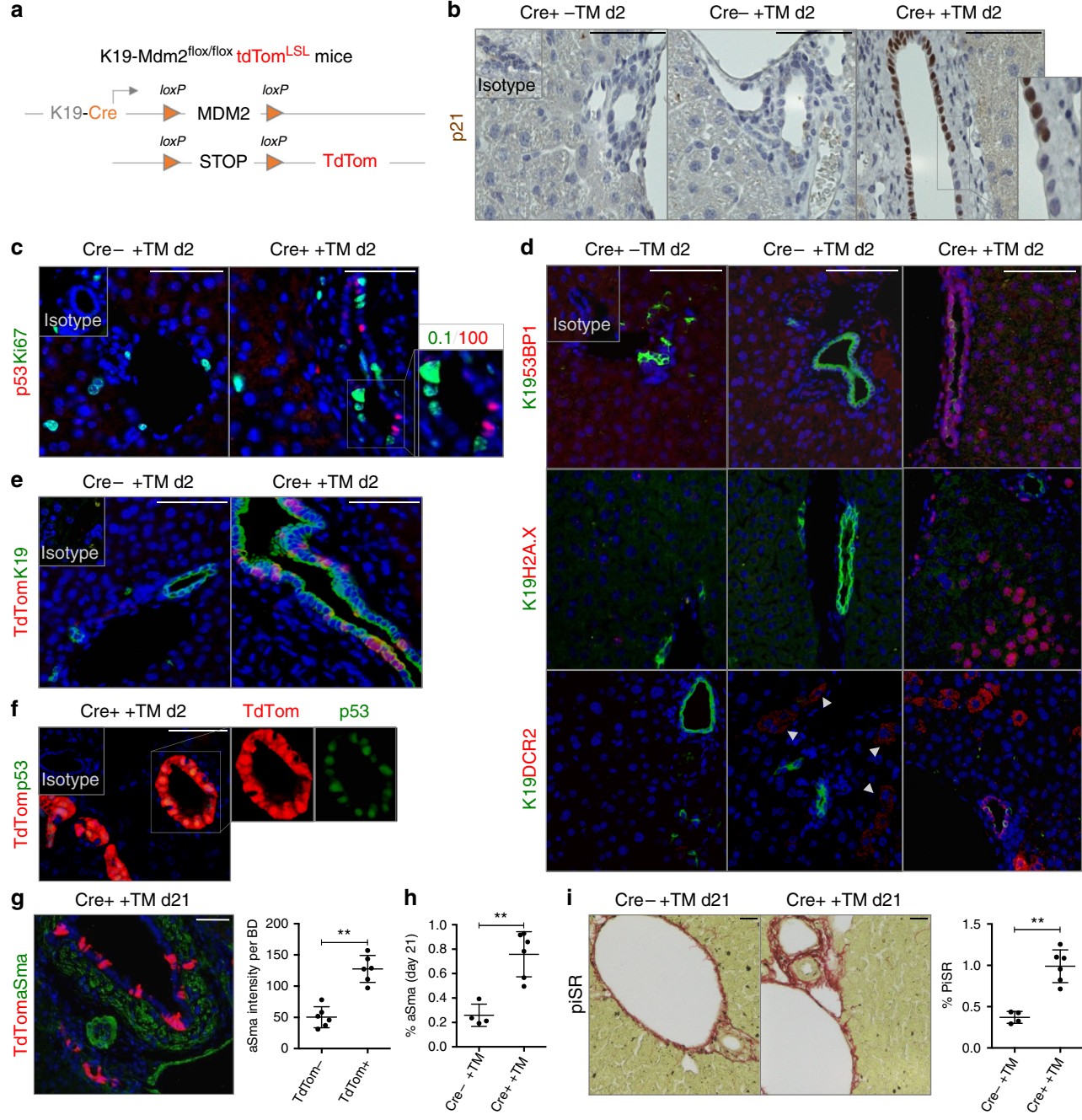

**Fig. 2** Cholangiocyte-specific deletion of *Mdm2* results in p21-driven cellular senescence. **a** Schematic explanation of the model: K19CreER[T] promoter is used to flox out *Mdm2* in cholangiocytes. Upon tamoxifen administration, Cre recombinase is activated and *Mdm2* is cleaved. The STOP sequence upstream tdTom reporter sequence is also floxed out making the identification of primary senescent events traceable through tdTom expression. **b** Increased levels of p21 in bile ducts 2 days after tamoxifen administration. Mouse data is presented at day 2 after final tamoxifen administration ($N = 6$ per group). (Left) K19Cre-positive mice without tamoxifen injection. (Center) K19Cre-negative mice after tamoxifen injection. (Right) K19Cre-positive mice after tamoxifen injection. **c** Cell-cycle-arrested cells do not proliferate. 0.1% (Mean ± SEM) of cholangiocytes co-express p53 (red) and Ki67 (green) 2 days after tamoxifen administration. **d** From top to bottom: 53BP1, γH2A.X and DCR2 (red) in K19-positive cholangiocytes (green). Notice the presence of 53BP1-positive, γH2A.X-positive and DCR2-positive hepatocytes in Cre+ +TM group. Bottom panel: Cre− +TM shows some DCR-positive hepatocytes (white arrowheads). **e** Recombined K19-positive cholangiocytes (green) can be traced by means of tdTom expression (red). **f** tdTom (red) co-localizes with p53 (green) in bile ducts. **g** Increased intensity of αSMA-positive cells (green) contiguous to tdTom-positive senescent-cholangiocytes (red) compared with tdTom-negative bile ducts at day 21 after induction $N = 6$. For an explanation about how this result was acquired, refer to Supplementary Fig. 5. **h** Increased total percentage of αSMA-positive cells in the Cre+ +TM group at day 21. **i** Increased deposition of collagen measured by PicroSirius Red (PiSR) at 21 days after tamoxifen induction. Right, quantification. ** denotes $p < 0.01$ (Mean ± SEM), Student's *t*test. Scale bars = 50 μm

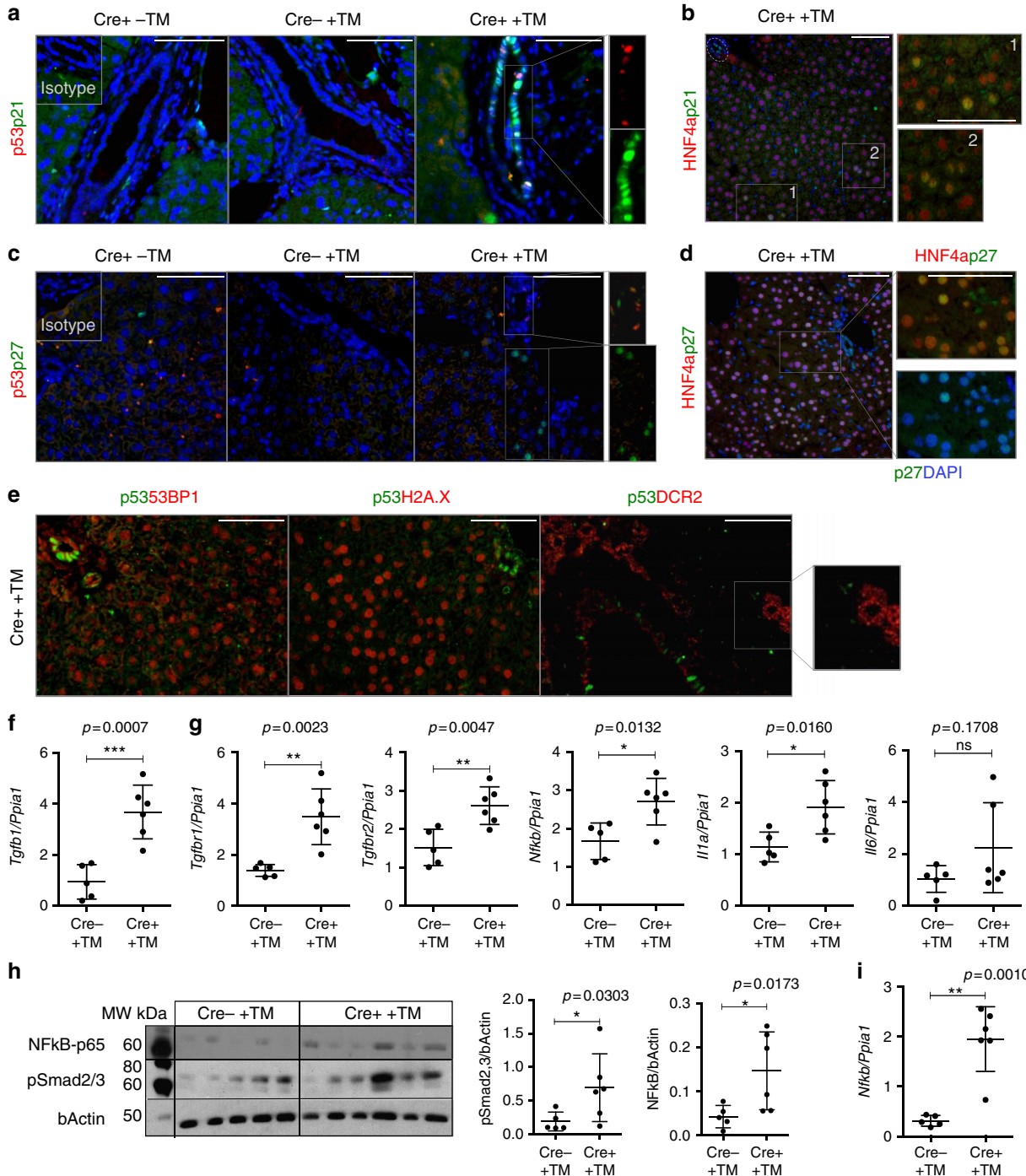

**Fig. 3** Paracrine senescence in the model is TGFβ-dependent. Mouse data is presented at day 2 after final tamoxifen administration (*N* = 6 per group). (Left) K19Cre-positive mice without tamoxifen injection. (Center) K19Cre-negative mice after tamoxifen injection. (Right) K19Cre-positive mice after tamoxifen injection. **a** p53 (red, indicative of primary senescence events in the cholangiocytes) and p21 (green) co-localize in the bile ducts. However, p21-positive p53-negative cholangiocytes were observed. **b** p21 (green) co-localizes with HNF4α-positive hepatocytes (red). **c** p27 (green) colocalizes with p53-positive cholangiocytes (red). p27 is expressed in cholangiocytes and hepatocytes in a paracrine manner. **d** p27 (green) co-localizes with HNF4α-positive hepatocytes (red). **e** Markers of senescence (such as 53BP1, H2A.X and DCR2, in red) are expressed in cholangiocytes and liver parenchyma, while expression of p53 (green) is restricted to the cholangiocytes. Single channels for this figure are provided in Supplementary Fig. 8a. Scale bars = 50 μm. **f** qRT-PCR analysis of *Tgfb1* in the isolated bile ducts of Cre− +TM (*N* = 5) and induced Cre+ +TM (*N* = 6) K19-Mdm2$^{flox/flox}$tdTom$^{LSL}$ mice day 2 after induction. *** denotes *p* < 0.005 (Mean ± SEM). Mann−Whitney test. **g** Analysis of common SASP's factors by qRT-PCR (from left to right, *Tgfbr1*, *Tgfbr2*, *Nfkb*, *Il1a* and *Il6*). * denotes *p* < 0.05, ** *p* < 0.01 (Mean ± SEM). Mann−Whitney test. **h** Western blot of phospho-NF-κB-p65 and pSmad2/3. Each band represents the bile ducts isolated from one mouse. Blots are companied by at least one marker position (molecular weights MW, in kDa). Far right, western blot densitometry quantification (normalized to bActin control) show increased expression of NF-κBp-65 and pSmad2/3 after induction of the model. * denotes *p* < 0.05 (Mean ± SEM). Mann−Whitney test. **i** qRT-PCR of *Nfkb* in hepatocytes isolated from Cre− and Cre+ K19-Mdm2$^{flox/flox}$tdTom$^{LSL}$ mice at day 2 after last tamoxifen administration (*N* = 5-6 per group). ** denotes *p* < 0.01 (Mean ± SEM). Mann−Whitney test

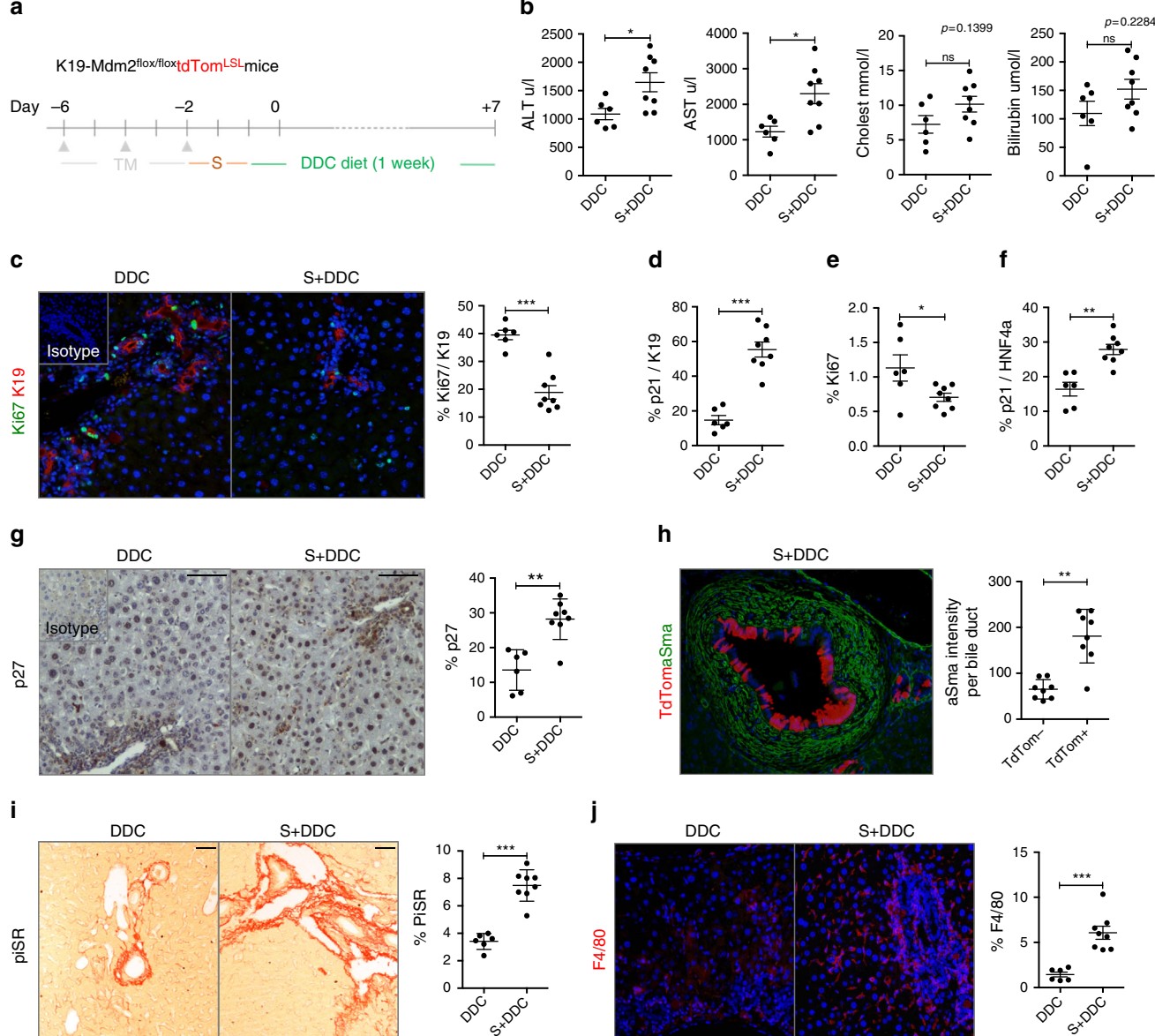

**Fig. 4** Cellular senescence in cholangiocytes aggravates biliary injury. **a** Schematic representation of the experiment. Two days after the induction of the model the DDC diet was administered for 1 week. Experimental groups include: DDC diet control group (DDC, where mice are administered with oil, $N = 6$) and Senescence + DDC diet (S + DDC, where animals receive tamoxifen, $N = 8$). **b** Serum analysis shows increased liver damage in the presence of senescent cholangiocytes. **c** Ki67-proliferating (green) cholangiocytes (red) decrease in the S + DDC group. **d** Increased number of p21-positive cholangiocytes in the S + DDC group. **e** Total percentage of proliferating cells decrease in the S + DDC group. **f** Increased number of p21-positive HNF4α-positive hepatocytes in the S + DDC group. **g** Increase of total number of p27 cells in the S + DDC group. **h** Increased presence of αSMA-positive cells (green) in the proximity of tdTom-positive senescent cholangiocytes (red). **i** Increased collagen deposition in the S + DDC group. **j** Increased number of F4/80-positive macrophages (red) in the S + DDC group. * denotes $p < 0.05$, ** $p < 0.01$, *** $p < 0.001$. (Mean ± SEM), Student's $t$test. Scale bars = 50 μm

cholangiocytes (Supplementary Fig. 7d). Furthermore, while expression of the tdTom or YFP is restricted to cholangiocytes, scattered expression of p21 can be seen in the parenchyma (Supplementary Fig. 7e), suggesting that there is a mechanism that induces senescence in a paracrine manner. Furthermore, we found p21-positive, p53-negative cholangiocytes (Supplementary Fig. 7f) in both models, suggesting a non-autonomous activation of p21, independent of the activation of p53 after Cre recombination.

Altogether, these results partially resemble the main characteristics of clinical PSC/PBC[1–3] and suggest that senescent cholangiocytes generate a local microenvironment that induces paracrine senescence in the surrounding hepatocytes, promotes macrophages recruitment and induces collagen deposition.

**Paracrine senescence in the model is TGFβ-dependent.** Following Cre activation, p21-positive, p53-negative cholangiocytes were observed in bile ducts (Fig. 3a). Furthermore, while p53 expression is restricted to cholangiocytes (Fig. 3a), we detected p21-positive HNF4α-positive hepatocytes (Fig. 3b).

p27, a direct target of transforming growth factor β (TGFβ)[11,19–21], is also expressed in cholangiocytes and hepatocytes in a similar pattern to p21 (Fig. 3c, d).

These senescent hepatocytes also express other senescence markers (53BP1, γH2A.X and DCR2), while p53 is restricted to the bile ducts (Fig. 3e and Supplementary Fig. 8a and b). These observations indicate that senescence triggered in hepatocytes is non-autonomous and p53-independent. This suggests the

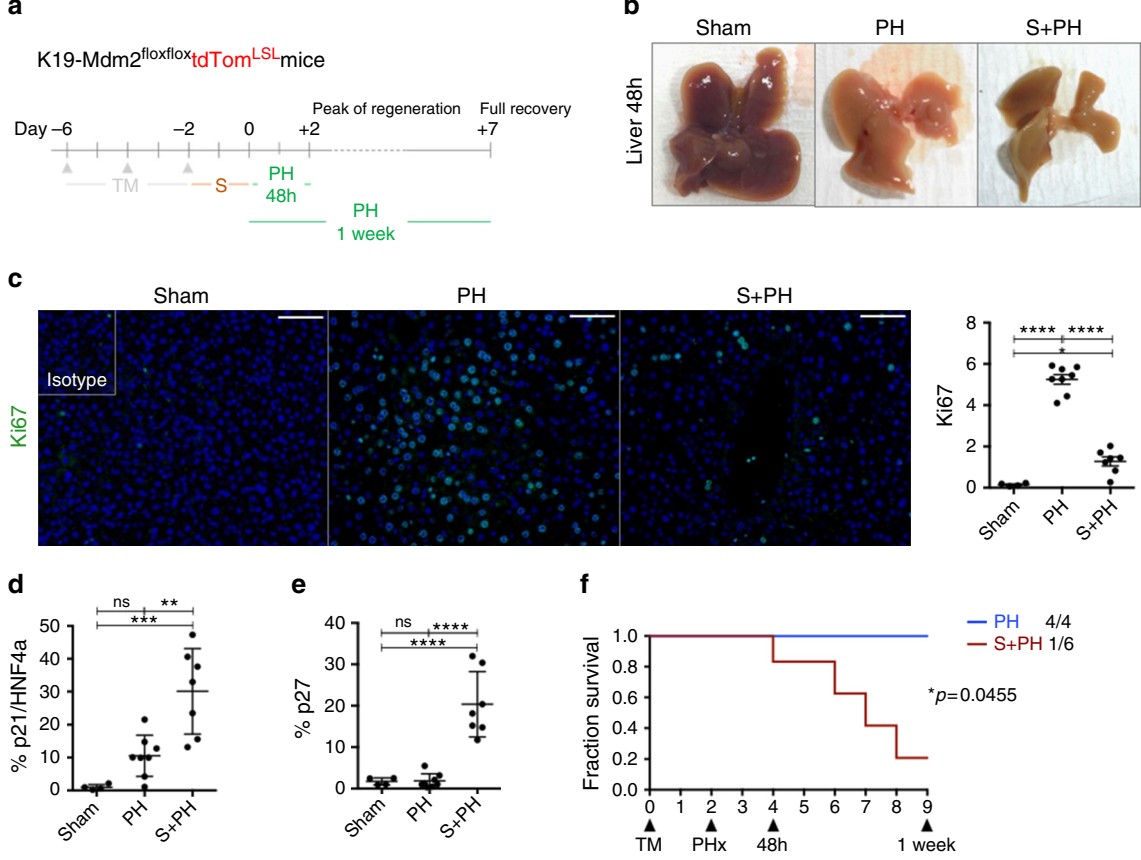

**Fig. 5** Cellular senescence in cholangiocytes impairs the regenerative response of liver parenchyma. **a** Schematic representation of the experiment. After induction of the model, we waited 2 days for display of senescence and performed 70% PH. Livers were recovered at day 2 after the surgery (peak of DNA replication in hepatocytes) and day 7 (when the liver mass is restored). Two groups were included in this experiment; PH ($N = 8$) and Senescence + PH (S + PH, $N = 7$). Sham animals are included as controls of the PH technique ($N = 4$). **b** Whole livers collected at 48 h post PH. From left to right: Sham (no manipulation of the liver), PH (which shows normal regenerative process by compensatory hypertrophy) and S + PH. Notice the absence of compensatory hypertrophy in the S + PH liver. **c** Decrease of Ki67 (green) in the S + PH group in comparison with PH group and Sham at 48 h post PH. Far right, quantification of total proliferation in the liver. *$p < 0.05$, ****$p < 0.0001$, (Mean ± SEM). ANOVA, Sidak's multiple comparisons test. Scale bars = 50 μm. **d** Increase of p21-positive hepatocytes in S + PH group at 48 h post-PH. **$p < 0.01$, ***$p < 0.001$ (Mean ± SEM). ANOVA, Sidak's multiple comparisons test. **e** Increase of total p27 in S + PH group at 48 h post-PH. ****$p < 0.0001$ (Mean ± SEM). ANOVA, Sidak's multiple comparisons test. **f** Survival curve for PH vs. S + PH at 1 week show a significant decrease of survival for S + PH mice (PH, $N = 4$; S + PH, $N = 6$). Log-Rank (Mantel−Cox test)

presence of a cholangiocyte-derived paracrine mediator that targets the liver parenchyma. To test this hypothesis, we focused into SASP signalling.

One of the most notable components of the SASP response is TGFβ. TGFβ can induce senescence in a paracrine manner, via a mechanism that generates reactive oxygen species and DDR[11,19–21,24,25]. TGFβ blocks the progression of the cell cycle at the restriction point in late G1 phase by transcriptional activation of cyclin-dependent kinase inhibitors (p21, p27 and p15,) through the SMAD complex[11,19–21,24,25].

We isolated bile ducts and hepatocytes from K19-Mdm2[flox/flox] tdTom[LSL] mice after the induction of senescence. As expected, isolated bile ducts express high levels of *K19* and low levels of *HNF4a*, as opposed to the hepatocytes (Supplementary Fig. 9a). Furthermore, bile ducts from induced mice present significantly increased mRNAs levels of *cdkn1a* (p21), *cdkn1b* (p27) and *cdkn2a* (p16) (Supplementary Fig. 9b).

The mRNAs encoding SASP proteins revealed a significant increased expression of *Tgfb1* in bile ducts ($p = 0.0007$) (Fig. 3f).

After senescence, isolated bile ducts also exhibit increased gene expression of *Tgfbr1* ($p = 0.0023$) and *Tgfbr2* ($p = 0.0047$) (Fig. 3g). Other SASP's factors were also significantly increased

in bile ducts, such as *Nfkb* ($p = 0.0132$) and *Il1a* ($p = 0.0160$) while there is a trend to increase in *Il6* level ($p = 0.1708$) (Fig. 3g and Supplementary Fig. 9c).

Western blot analysis also revealed increased protein levels of pSmad2/3, suggesting a direct functional role for TGFβ in the activation of the SMAD complex (Fig. 3h).

Increased levels of phospho-NF-κB-p65, a master regulator of the SASP[26], at protein (Fig. 3h) and gene level (see Fig. 3g) suggest a robust activation of the SASP in bile ducts. Interestingly, we observed upregulation of *Nfkb* in hepatocytes after Cre activation ($p = 0.0010$) (Fig. 3i). Isolated hepatocytes also significantly upregulate *Tgfbr1* ($p = 0.0043$) and *Tgfbr2* ($p = 0.0173$) (Supplementary Fig. 9d).

Furthermore, in situ hybridization (ISH) for *Tgfb1*-ligand also suggests that TGFb is produced by cholangiocytes and surrounding hepatocytes (Supplementary Fig. 9e). These results suggest a potential mechanism of paracrine senescence activation triggered by the SASP in the parenchyma.

The above observations indicate that the K19-Mdm2[flox/flox] tdTom[LSL] model robustly establishes senescence in cholangiocytes and induces a TGFβ-dependent SASP response in the parenchyma.

**Cellular senescence aggravates biliary injury**. To test if cholangiocyte senescence affects the outcome of biliary injury, we administered 1 week of diethoxycarbonyl-1,4-dihydro-collidine (DDC) diet to the K19-Mdm2$^{flox/flox}$tdTom$^{LSL}$ model (Fig. 4a). The DDC diet has been previously used to induce features of PSC/PBC[27] but, up to date, the effects of senescence in this model are not known.

Following DDC administration, there is a significant increase of serum transaminases in the tamoxifen-injected group (Fig. 4b). Decrease in cholangiocyte proliferation (Fig. 4c) and p21-positive cholangiocytes (Fig. 4d) can be observed in concurrence with decreased total liver proliferation (Fig. 4e). Increased number of p21-positive hepatocytes (Fig. 4f) and p27-expressing cells can be seen in the tamoxifen-injected group

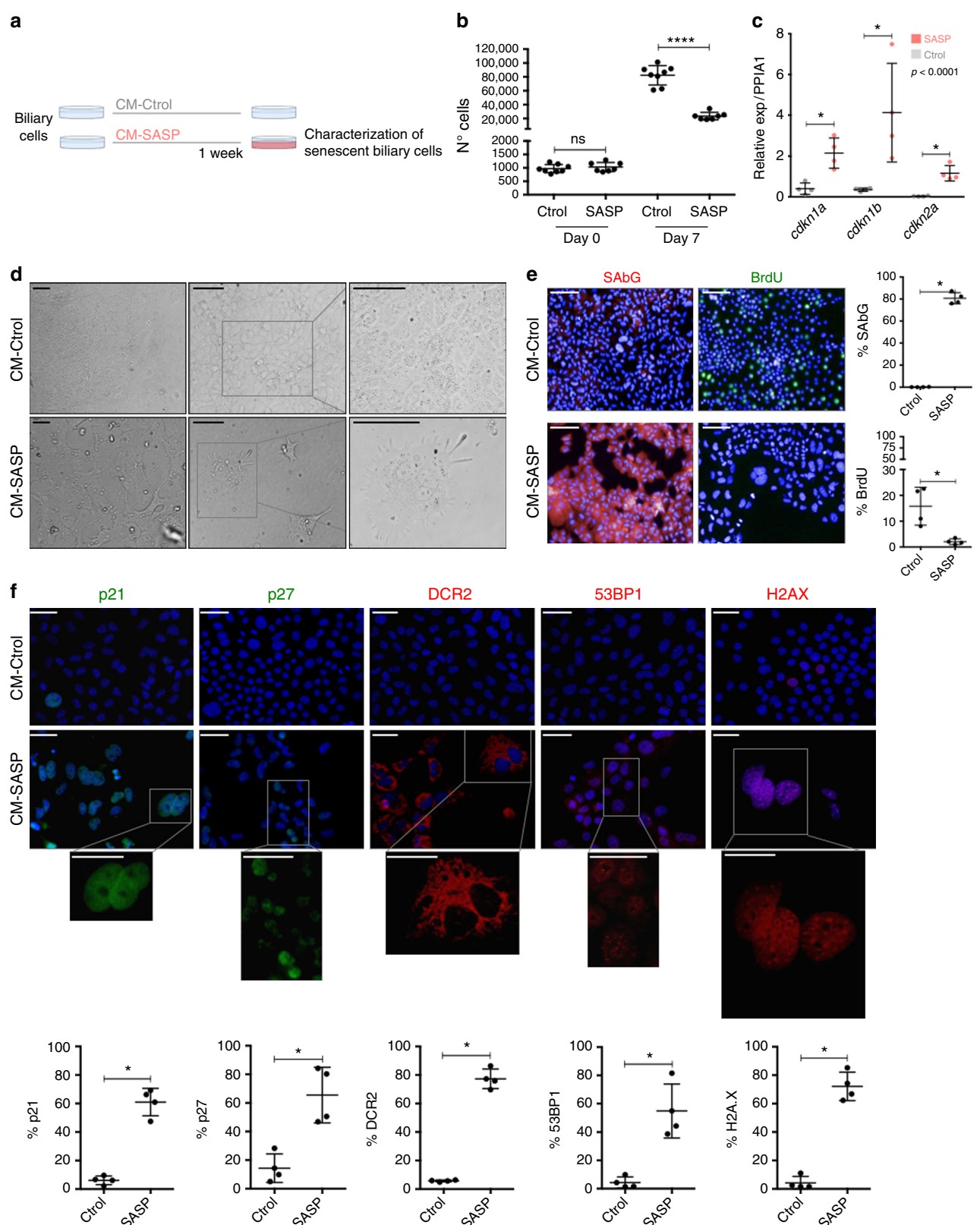

(Fig. 4g), supporting the hypothesis that senescence spreads from cholangiocytes to the rest of the liver. Further comparison between control groups (healthy wild-type mice, wild-type mice on DDC diet and the uninduced K19-Mdm2$^{flox/flox}$tdTom$^{LSL}$ model) can be found in Supplementary Fig. 10a–e.

Furthermore, increased αSMA-positive cells proximal to tdTom-positive cholangiocytes (Fig. 4h) and collagen deposition (Fig. 4i) suggest that senescent cells generate a local microenvironment that results in fibrogenesis. Increased number of F4/80-positive macrophages (Fig. 4j) can also be observed in the tamoxifen-injected group. We did not find significant differences in the total number of K19-positive cells between the vehicle and tamoxifen-injected group after DDC diet (Supplementary Fig. 10f).

These data altogether suggest that senescence can aggravate biliary injury by spreading to the parenchyma.

**Cellular senescence impairs liver parenchyma regeneration.** To test whether the effects of paracrine senescence upon the hepatocytes affect the regenerative capacity of the liver, we induced senescence in cholangiocytes in the K19-Mdm2$^{flox/flox}$tdTom$^{LSL}$ model and tested hepatocyte regenerative capacity with 70% partial hepatectomy (PH), a well-described model of hepatocyte-mediated liver regeneration (Fig. 5a)[28].

In the livers subjected to senescence and PH, compensatory hypertrophy of the remaining lobes is reduced (Fig. 5b). Decreased proliferation (Fig. 5c and Supplementary Fig. 11a) and increased p21-positive hepatocytes (Fig. 5d and Supplementary Fig. 11b) occur in the senescent group. We observed increased p27 expression (Fig. 5e) suggesting an active role of TGFβ-dependent SASP mechanism in the parenchyma.

One week after PH there was a marked decrease in survival in the senescent group (Fig. 5f), suggesting that the livers subjected to cholangiocyte senescence and SASP effects are unable to compensate the loss of liver tissue. We did not find evidence of the involvement of mesothelial cells in our model (Supplementary Fig. 11d).

These results suggest that cholangiocyte senescence impairs the regenerative response of the liver parenchyma. This could potentially account for the loss of hepatocyte function seen in human PSC/PBC patients.

**Senescence is transmitted in vitro through TGFβ.** To test whether transmission of senescence from cholangiocytes to the surrounding hepatocytes is TGFβ-dependent, we generated an in vitro model based on the induction of paracrine senescence in culture expanded biliary cells derived from normal mouse liver as previously described[29].

As previously shown by Tabibian and collaborators, N-Ras activation is an intrinsic characteristic of senescent cholangiocytes in PSC[8]. We generated a system of paracrine senescence based upon N-RAS activation (Fig. 6). Briefly, we added conditioned media from senescent cells (CM-SASP) to our biliary cells

(Fig. 6a). CM-SASP is produced by IMR90 ER:RAS cells in which senescence is induced by means of the activation of a RAS-chimaeric fusion protein[12].

CM-SASP induced senescence in the biliary cells, causing growth arrest (Fig. 6b), with increased mRNA levels of *cdkn1a*, *cdkn1b* and *cdkn2a* (Fig. 6c). Altered cell morphology (Fig. 6d) and upregulated expression of several senescent markers with reduced BrdU incorporation (Fig. 6e, f) are observed.

With the same method, we induced senescence in biliary cells, withdrew the CM and then added YFP-positive biliary cells (Fig. 7a). This allows the assessment of the paracrine senescence transmission from the senescent biliary cells to the YFP-biliary cells. Quantifications in the YFP-biliary cells showed increased senescent markers and decreased BrdU incorporation, demonstrating that senescence can be transmitted in a paracrine manner in vitro (Fig. 7b).

In order to inhibit the transmission of paracrine senescence, we blocked TGFβ signalling pathway and assessed the same markers of senescence in the YFP-biliary cells. To that purpose, we used a small molecule that targets the TGFβ receptors ALK4, ALK5 and ALK7[30] (LY2157299), effectively blocking TGFβ signalling pathway (Fig. 7c).

Addition of LY2157299 caused a significant reduction of paracrine senescence in the YFP-biliary cells (Fig. 7d and Supplementary Fig. 12a, b), without altering the number of primary senescent biliary cells (Supplementary Fig. 12c). These results accentuate the importance of TGFβ in the transmission of paracrine senescence, and offers a potential approach for the inhibition of paracrine senescence in biliary disease.

Finally, we investigated whether macrophages are able to clear senescent biliary cells. We established an in vitro system in which senescent YFP-biliary cells were co-cultured with bone marrow-derived macrophages artificially labelled with 647-Cell Mask (647-BMDM). Interaction between the senescent YFP-biliary cells and 647-BMDM can be observed in the co-culture system (Supplementary Fig. 13a–h). The number of senescent biliary cells significantly decreased after 72 h when cultured with 647-BMDM (Supplementary Fig. 13h), suggesting that BMDMs can target and eliminate senescent biliary cells.

**Inhibition of TGFβ in vivo improves liver function.** We investigated the therapeutic effect of TGFβ signalling disruption during paracrine senescence in the DDC biliary disease model.

LY2157299 treatment (Fig. 8a) resulted in decreased number of p21-positive cholangiocytes (Fig. 8b, c), a significant reduction of total p21 (Fig. 8d) and p27-expressing cells in the liver (Fig. 8e), reflecting a decrease in the paracrine senescence response. The number of primary senescent cholangiocytes (tdTom-positive) remain unaltered (Fig. 8f), proving that primary senescence is not affected by the administration of LY2157299.

Administration of LY2157299 in the K19-Mdm2$^{flox/flox}$td-Tom$^{LSL}$ model prior to DDC diet (Fig. 8g) resulted in improved liver function as shown by decreased serum transaminases

**Fig. 6** CM-SASP induces senescence in a paracrine manner. **a** Schematic representation of the experiment. Biliary cells are cultured with SASP-conditioned media (CM-SASP) and Control-conditioned media (CM-Ctrol). After 1 week, different markers of senescence were evaluated. **b** Number of total Ctrol-treated biliary cells increase (suggesting that they continue proliferating). Total number of CM-SASP-treated biliary cells *plateau*, indicative of an impaired proliferative response. **** denotes $p < 0.0001$ (Mean ± SEM). Mann−Whitney test. $N = 7$–8 biological replicates. **c** qRT-PCR shows a significant increased expression of *cdkn1a* (p21), *cdkn1b* (p27) and *cdkn2a* (p16) in the CM-SASP-treated biliary cells at day 7. * denotes $p < 0.05$ (Mean ± SEM). Mann−Whitney test. $N = 4$ biological replicates. **d** Morphological changes observed in the cell cultures. CM-SASP-treated biliary cells become flatter, larger and their content is vacuolized. **e** Increased SA-βGal (red) expression in CM-SASP-treated biliary cells with decreased BrdU incorporation (green). Far right, total percentage of SA-βGal or BrdU per DAPI-positive nuclei per field. **f** Increased expression of different senescence markers in CM-SASP-treated biliary cells compared with the CM-Ctrol-treated biliary cells. Below, total percentage of those markers per DAPI-positive nuclei. * denotes $p < 0.05$ (Mean ± SEM). Mann−Whitney test. $N = 4$ biological replicates. Scale bars = 50 μm

(Fig. 8h), increased number of proliferating cholangiocytes (Fig. 8i) and increased liver proliferation (Fig. 8j). These results correlate with a trend towards decrease in p27-positive cells ($p = 0.0952$; Fig. 8k), and reduced p21-positive and p16-positive hepatocytes (Fig. 8l). We also observed a decrease in collagen deposition (Fig. 8m) that may reflect a decrease in the fibrogenic

response, and a trend towards decreased numbers of F4/80 macrophages ($p = 0.1508$; Fig. 8n).

## Discussion

Cellular senescence is the permanent cell-cycle arrest related to the process of ageing[31] and has recently been described as a

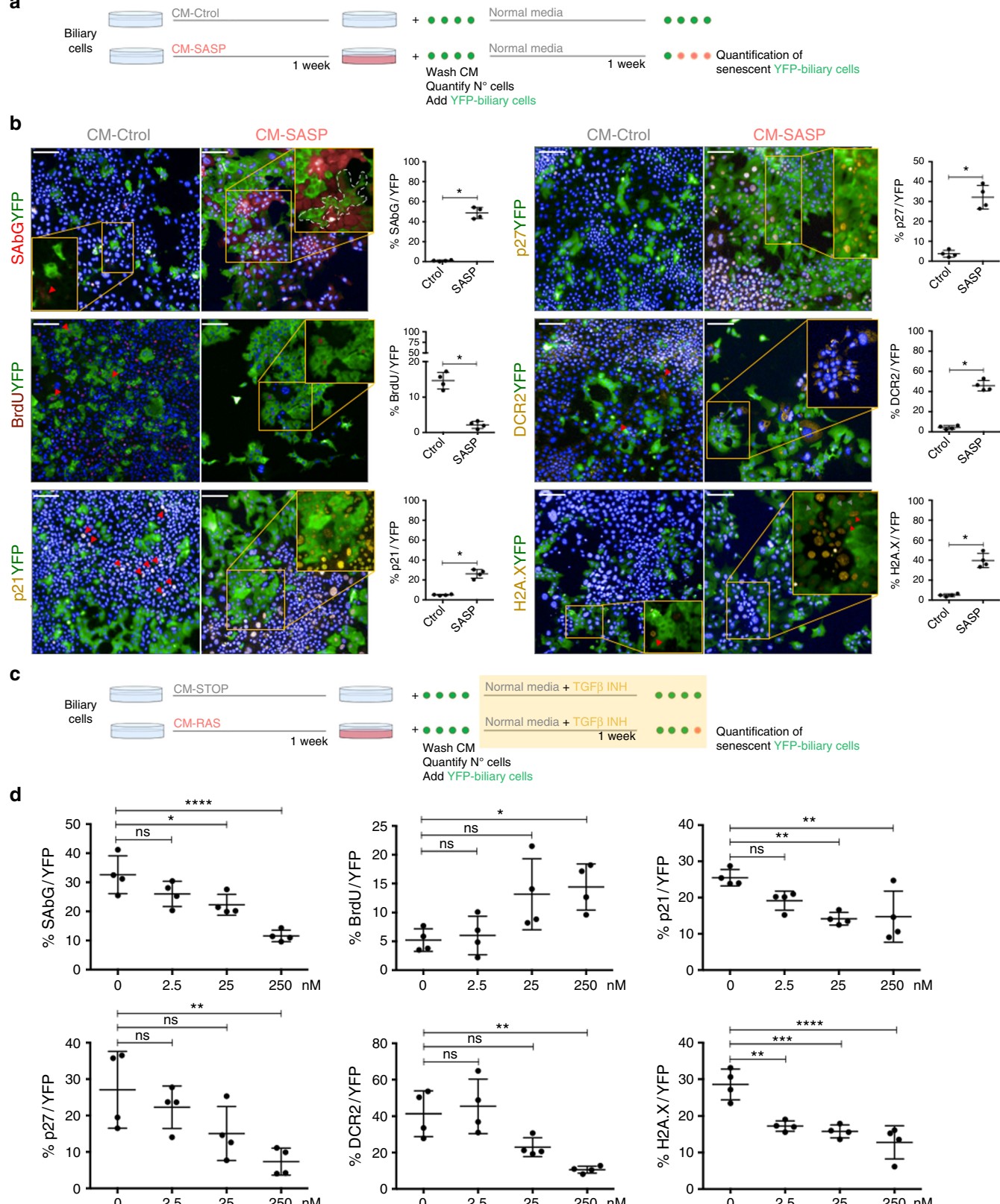

tumour-preventive mechanism[32]. In the liver, it has been associated with cholangiocytes in end-stage liver disease[33], identified in hepatocytes in cirrhotic livers[34,35], and described as a mechanism to control hepatic stellate cell-mediated fibrogenesis[13].

In biliary disease, senescence has proved to be an essential characteristic of PSC/PBC: Sasaki et al. described the presence of senescent cells in human PSC/PBC[5,6], and Tabibian et al. suggested the importance of N-Ras-induced senescence and its paracrine effects in PSC[8]. Furthermore, the *Mdr2* knock-out mouse model exhibits high levels of p16-dependent cholangiocyte senescence[36], and is characterized by the spontaneous development of cholangitis and 'onion skin' periductal fibrosis, suggesting a potential contribution of senescence to the development of biliary disease. These observations highlight the importance of cholangiocyte-senescence in biliary disease.

We have developed a highly efficient model to induce senescence specifically in cholangiocytes, partially resembling human biliary disease. This model offers the advantages of conditional activation and can be used as a tool to study the mechanisms of biliary senescence in the liver. The presence of senescent cholangiocytes in this model promotes ductular reaction, increases deposition of collagen and impairs liver regeneration after injury. The presence of αSMA-positive cells in the proximities of the senescent cholangiocytes suggests that senescence might have a role in the development of fibrosis, characteristic of human PSC/PBC.

After induction, our model exhibits bile duct apoptosis. This suggests that biliary senescence generates an environment permissive for the initiation of apoptotic mechanisms (previously described as a potential mediator of ductopenia[4]), but further investigations are required to investigate the mechanisms behind this phenomenon.

The presence of senescent cholangiocytes aggravates biliary disease upon administration of DDC diet and confirms that senescent cholangiocytes exert a detrimental role in the development of biliary disease and regeneration. Senescent cholangiocytes affect their microenvironment, altering macrophage infiltration, collagen deposition and impairing biliary functions. The induction of senescence in cholangiocytes also results in the establishment of paracrine senescence in the liver parenchyma through TGFβ-dependent mechanisms. This suggests that the initial wave of senescence in the bile ducts might spread and compromise the regenerative ability of the liver parenchyma, confirmed by the decreased survival of animals subjected to biliary senescence and 70% PH.

As a role for senescent cholangiocytes in the attraction of macrophages has been suggested in PBC[6], we used an in vitro model to investigate the interaction between macrophage infiltration and senescent cholangiocytes observed in the in vivo model. The number of senescent cholangiocytes decreases when co-cultured with BMDM, suggesting that macrophages can target and eliminate senescent cholangiocytes. These observations provide a potential mechanism that explains the disappearance of bile ducts seen in human PSC/PBC[4,6]. Destruction of biliary architecture is not present in this model; however, ductopenia is a late feature in PSC/PBC[4,5] and likely occurs over years of injury and impaired regeneration, which is challenging to simulate in a mouse model.

Inhibition of TGFβ signalling partially blocks paracrine senescence effects. Consistently, we have demonstrated that in vivo partial ablation of the SASP through TGFβ inhibition decreases paracrine senescence, increases hepatocyte proliferation and liver function. Furthermore, it appears to decrease fibrogenesis in our model (although it should be noted that TGFβ inhibition can also decrease liver fibrosis per se[37]).

We suggest that senescence may contribute to the development of biliary disease through complementary mechanisms. First, through an impaired regenerative response of cholangiocytes[38], unable to compensate biliary damage, and second, through SASP expression, that can induce paracrine senescence in hepatocytes (thus diminishing the regenerative capacity of the liver during injury)[11,12], promote collagen deposition and enhance fibrogenesis[34,35].

Overall, we have shown that cellular senescence is likely to be a driver of biliary injury by affecting the microenvironment, impairing liver parenchyma regeneration and impairing biliary function. Importantly, we have found that this pathology can be specifically targeted through TGFβ inhibition.

## Methods

**Clinical material**. Human liver biopsies from a clinical series of PSC and PBC cases (paraffin-embedded samples, $n = 7$ per group) were anonymized and assessed histologically using H&E by an expert pathologist. Diagnoses were based on clinical data and confirmed by histology (see Supplementary Table 1). Normal human liver samples were obtained from autopsy cases from non-hepatic disease with normal liver histology ($n = 7$). Informed consent was obtained from the patients for the tissue to be used in the Tissue Governance Group, College of Medicine and Veterinary NHS Lothian Public Health Office, under project license TGU-LAB-SR299 approved by the Local Commission for Medical Ethics, University of Edinburgh.

**Animal models**. Mice were housed in a specific pathogen-free environment in open cages (NKP, M3-sloping front) with Aspen chips as bedding at 21 °C. Mice were kept under standard conditions with a 14-h day cycle and access to food (irradiated RM3P) and water ad libitum. Power calculations are not routinely performed. However, animal numbers were chosen to reflect the expected magnitude of response taking into account the variability observed in previous experiments. Mice were randomly allocated to each experimental group and males/females are equally distributed. All animal experiments were carried out under procedural guidelines, severity protocols and within the UK with ethical permission from the Animal Welfare and Ethical Review Body (AWERB) and the Home Office (UK).

K19CreER$^T$+ and K19CreER$^T$− mice were crossed with Mdm2$^{flox/flox}$ and R26RtdTomato$^{LSL}$ mice to generate K19-Mdm2$^{flox/flox}$tdTom$^{LSL}$ line (Cre+ and Cre−). The animals used in this study are C57Bl/6 background, mix of males and females aged within 8–12 weeks at the start of the experiments.

C57Bl/6 mice were used in some figures (see as example Supplementary Fig. 10) as healthy control. C57Bl/6 mice under 1 week of DDC diet were also included as control of the DDC diet in our model.

OPNiCreER$^{T2}$+ and OPNiCreER$^{T2}$− mice were crossed with Mdm2$^{flox/flox}$ and R26R$^{YFP}$ mice to generate OPN-Mdm2$^{flox/flox}$YFP line (Cre+ and Cre−). The animals used on this study have CD1-enriched background, mix of males and females aged within 8–12 weeks at the start of the experiments.

K19CreER$^T$ and R26RtdTomato$^{LSL}$ were purchased from Jackson Laboratory.

**Fig. 7** Cellular senescence can be transmitted in vitro. Inhibition of TGFβ decreases paracrine senescence. **a** Experimental scheme; biliary cells were treated with RAS-conditional media (CM-RAS) and CONTROL-conditional media (CM-Ctrol) for a week. Media is eliminated and cells washed with PBS five times. YFP-positive biliary cells (YFP-biliary cells) are then added and after 1 week different markers of senescence in the YFP-biliary cells population were assessed. **b** Representative images of the assessed markers of senescence and proliferation (BrdU) in the co-culture of senescent biliary cells (YFP-negative) and YFP-positive cells. Data is presented as biliary cells treated with CM-STOP and CM-RAS at final time point (14 days). Far right, percentage of each marker in the YFP-biliary cells population. * denotes $p < 0.05$ (Mean ± SEM). Mann−Whitney test. $N = 4$ biological replicates. Scale bars = 50 μm. **c** Experimental scheme for the use of TGFβ inhibitor (LY-2157299); YFP-biliary cells and different concentrations of LY-2157299 are added to the senescent biliary cells. After 1 week different markers of senescence in the YFP-biliary cells population were assessed. **d** Decrease of senescence factors and increase in proliferation in YFP-biliary cells with increasing concentrations of LY-2157299. * denotes $p < 0.05$, **$p < 0.01$, ***$p < 0.001$, ****$p < 0.0001$ (Mean ± SEM). ANOVA, Sidak's multiple comparisons test. $N = 4$ biological replicates

OPNiCreER$^{T2}$R26R$^{YFP}$ mouse line was a kind gift by F. Lemaigre and I. Leclercq, Institut de Recherche Expérimentale et Clinique and Duve Institute Brussels, Belgium. Mdm2$^{flox/flox}$ mice were generously provided by Professor Gigi Lozano, Department of Cancer Genetics, University of Texas. All animal genotyping was outsourced commercially to Transnetyx, Inc (TN, USA) using the primers stated in Supplementary Table 2.

Further details (ARRIVE guidelines) about the use of the animal models can be found in Supplementary Note 1.

**Induction of the model and recovery of samples.** Recombination of loxP sites was induced with three doses of 4 mg tamoxifen (Sigma) in sunflower seed oil (Sigma) by intraperitoneal injection on alternate days. Mice were killed according to UK Home Office regulations and blood collected by cardiac puncture and centrifuged to collect serum. Livers were perfused in situ with PBS through the portal vein and harvested. Organs were harvested and either directly frozen at −80 °C or fixed in 10% formalin (in PBS) for 12 h prior to embedding in paraffin blocks.

**DDC diet.** To induce biliary injury, mice were given 0.1% 3,5-diethoxycarbonyl-1,4-dihydrocollidine (DDC) mixed with Rat & Mouse No Maintenance (RM1) diet (Special Diet Services) as previously described[39]. DDC diet was administered for 7 days, 2 days after last tamoxifen injection. Although the literature describes prolonged periods of DDC feeding in different mouse models, the

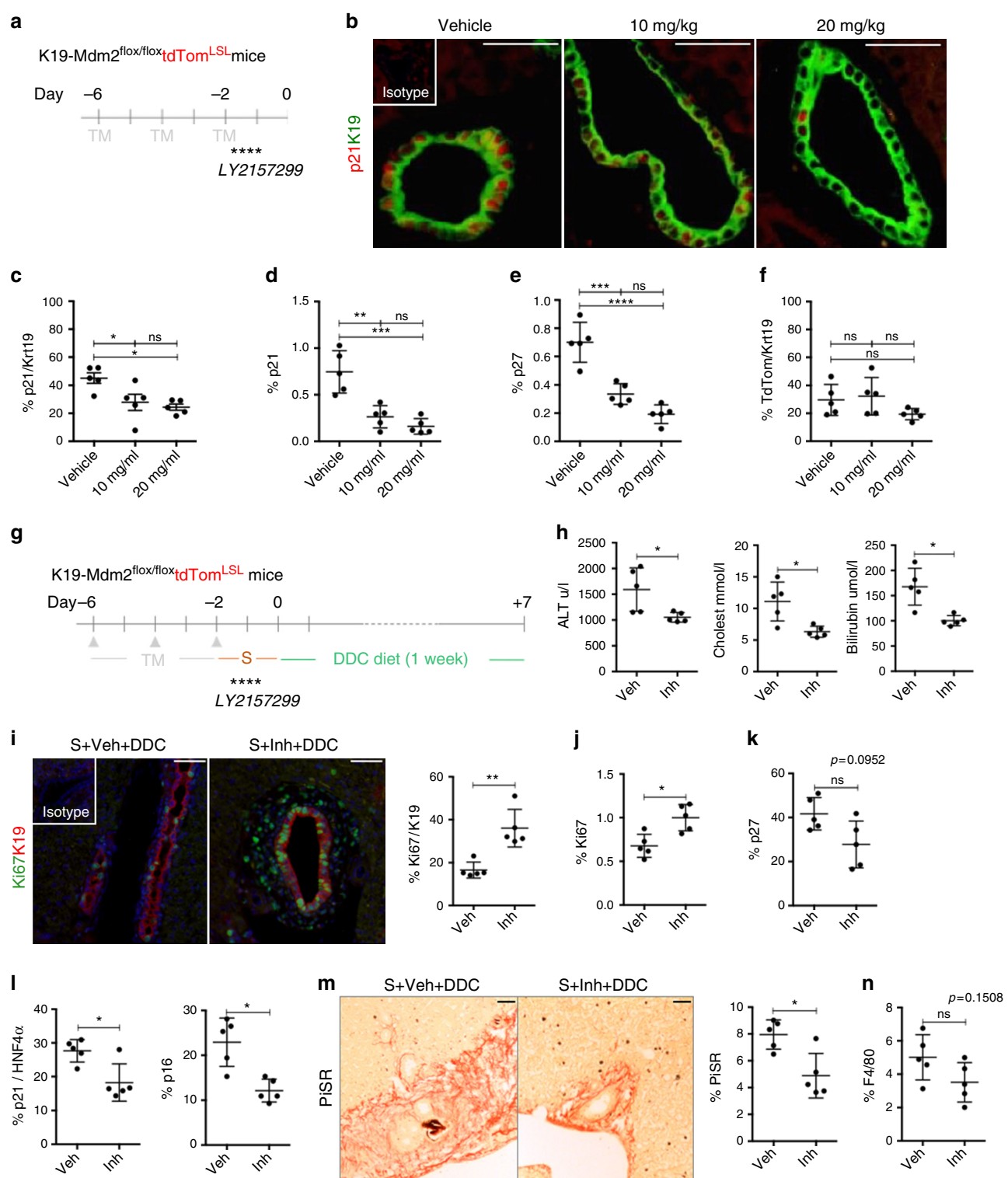

K19-Mdm2$^{flox/flox}$tdTom line only admitted 1 week of DDC diet due to its C57BL6/J background. This decision takes into consideration restrictions of a maximum total weight loss of 20%.

**70% partial hepatectomy**. Briefly, median laparotomy was performed under isoflurane anaesthesia and left lateral and median lobe were removed as previously described[28]. To avoid damaging the gallbladder, the 'three knots' technique was performed. Sham-operated mice were subject to laparotomy and brief manipulation of the intestines, but no hepatectomy was performed. Mice were killed at 48 h and 1 week after surgery.

**In vivo use of TGFβR1 antagonists**. Galunisertib (LY2157299, Cayman Chemical Company) was administered at 10 mg/kg twice daily by gavage after the induction of the model until day 2, when we administered DDC diet as stated. LY2157299 was prepared in 10% polyethylene glycol, 5% DMSO and 85% saline vehicle. Control mice were administered only with the vehicle (PEG, DMSO, Saline).

**Serum analysis**. Serum analysis used commercial kits according to the manufacturer's instructions; serum albumin (Alb), bilirubin and alanine transaminase (ALT) (Alpha Laboratories); aspartate aminotransferase (AST) and alkaline phosphatase(ALP) (Randox laboratories); Cholesterol (Wako Chemicals GmbH). All kits were adapted for use on a Cobas Fara centrifugal analyser (Roche Diagnostics).

**Histology**. Paraffin sections (4 μm) were cut and stained. Antibodies used for antigen detection in the current study are provided together with method of antigen retrieval, working dilution and isotype control (Supplementary Table 3).

**Immunohistochemistry**. For tissue, 100 mM sodium citrate pH 6.0 or 100 mM Tris-EDTA pH 9.0 were used for antigen retrieval. Sections were blocked for endogenous peroxidase and alkaline phosphatase activity (Bloxall, Vector), endogenous avidin/biotin (Invitrogen) and nonspecific protein binding (Spring Bioscience). Primary antibodies were incubated as stated in Supplementary Table 3. Species-specific secondary biotinylated antibodies (Vector), Vectastain R.T.U, ABC reagent (Vector) and DAB chromogen (Dako) were used sequentially to detect the primary antibody. Sections were counterstained with haematoxylin.

Isotype control is used in the same concentration as the primary antibody. The sample is then incubated with the secondary antibody and detection reagents.

**Immunofluorescence**. For tissue, 100 mM sodium citrate pH 6.0 or 100 mM Tris-EDTA pH 9.0 were used for antigen retrieval. Sections were blocked with protein block (Spring Bioscience) and incubated using antibodies listed in Supplementary Table 3. Primary antibodies were detected using fluorescent-conjugated secondary antibodies (Alexa-488, 555 or 650, Invitrogen). Sections were mounted with DAPI containing media (DAPI Fluoromount-G, SouthernBiotech).

Isotype control is used in the same concentration as the primary antibody. The sample is then incubated with the secondary antibody and detection reagents.

**TUNEL**. Apoptosis was assessed using In Situ Cell Death Detection Kit (Roche) according to the manufacturers' instructions. Positive control was performed using 10 units/ml of DNAseI (Qiagen).

**Haematoxylin and eosin**. Sections were stained routinely and mounted in fluorescence-free DPX mountant (Sigma).

**PicroSirius Red**. Sections were stained using picric acid (P6744-1GA), fast green (F7258-25G) and direct red (365548-25G, all from Sigma Aldrich).

**In situ hybridization**. In situ mRNA hybridization (ISH) was performed using RNAscope LS probes for *Tgfβ1*, *Mdm2* and *PPIB* control (407758, 447648 and

313918; Advanced Cell Diagnostics) as per the manufacturer's instructions. Images were acquired and each probe was quantified in cholangiocytes and hepatocytes, which are identified based on their morphology.

**In vitro studies**. Biliary cells were obtained as previously described and routinely used in our laboratory[29,40]. YFP-Biliary cells is a stably transfected GFP biliary cell line derived in the laboratory[29]. Prior use in our experiments, YFP-Biliary cells were cultured with medium supplemented with 3 μg/ml of puromycin (Thermo Fisher) for 14 days to exclude cells that randomly lose YFP. YFP expression of stably transfected lines was constantly monitored with fluorescence microscopy.

Biliary cells are cultured in Williams' E 10% FCS supplemented with 20 mM HEPES pH 7.5, 17.6 mM NaHCO₃, 10 mM nicotinamide, 1 mM sodium pyruvate, 1× ITS, 100 nM dexamethasone, 0.2 mM ascorbic acid, 14 mM glucose 1.4 M, 10 ng/ml IL-6 10 μg/ml, 10 ng/ml HGF 20 μg/ml, 10 ng/ml EGF 100 μg/ml as previously described[40]. All experiments were performed at passage 16–17.

**Induction of senescence with conditioned medium**. To induce senescence, we treated biliary cells with conditioned media for a week. For co-culture experiments, cells were counted, conditioned media eliminated and cells washed with PBS1x five times before plating same number of YFP-Biliary cells. For the inhibition of TGFβ signalling experiment, LY2157299 (Cayman Chemical Company) solubilized in DMSO was added at the stated concentrations. DMSO was used for controls.

Conditioned media-SASP (CM-SASP) was produced as described in Acosta et al.[12]. Briefly, IMR90 ER:RAS cells (2×10⁶) were seeded in a 10 cm dish and incubated for 7 days with 200 nM 4OHT in DMEM with 0.5% FBS. After incubation, the conditioned medium (CM) was collected, centrifuged at $5000 \times g$ and filtered through a 0.2 μm pore filter. CM was mixed with DMEM 40% FBS in a proportion of 3 to 1 to generate CM containing 10% FBS. Control conditioned media (CM-Ctrl) is produced in the same manner from IMR90 ER:STOP cell line.

**DDAOG detection of SA β-Galactosidase activity**. Cells were sequentially cultured with 100 nM Bafilomycin A1 (Sigma) and 100 μM DDAOG (Thermo), 1 and 2 h respectively at 37 °C as previously described[41]. Cells were then fixed 15 min 4% paraformaldehyde and permeabilized with 0.2% Triton X-100 for 10 min. Images were acquired the same day at 647 nM exposure.

**5-Bromo-2'-deoxyuridine incorporation**. 5-Bromo-2'-deoxyuridine (BrdU) (Invitrogen) is added to the cells at 3 μg/ml and cultured 3 h at 37 °C. Cells were fixed in cold Methanol:Acetone (1:1) for 10 min and treated with 2 N HCl 30 min at RT. Immunofluorescence was routinely performed (Supplementary Table 3). Nuclear staining was performed with DAPI.

**Immunofluorescence for cells**. Cells were fixed in cold Methanol:Acetone (1:1) for 10 min and permeabilized with 0.1% Triton A-100 (Sigma) during 15 min before immunofluorescence was routinely performed. Antibodies used are listed in Supplementary Table 3. Nuclear staining was performed with DAPI.

**Microscopy and cell counting**. Images were acquired using a Nikon Eclipse e600 and Retiga 2000R camera (Q-Imaging, Image Pro premier software) or Perkin Elmer Operetta high content imaging system. For single-cell quantification in tissue, images were acquired in up to four fluorescent channels at ×20 magnification, and analysed using Fiji ImageJ or Columbus Image Data Storage and Analysis System (Perkin Elmer) software depending on the microscope used.

Histological sections were assigned a randomized blinded code prior to quantification by a separate researcher, and the randomization decoded at the time of the final data analysis. Quantification of different liver populations from the images was performed manually on 30 random non-overlapping fields per mouse (×200 magnification).

Ductopenic index was calculated by dividing the number of K19-positive bile ducts per portal tract. Any ratio below a 0.5 threshold was considered ductopenia[42].

Cells were identified based on DAPI stained-nuclei, morphology and specific staining for each population. Illumination correction and background

**Fig. 8** Use of TGFβ inhibitors in vivo impairs transmission of paracrine senescence and improves liver function. **a** Pattern of administration of LY2157299 by oral gavage after induction of senescence in the K19-Mdm2$^{flox/flox}$tdTom$^{LSL}$ model. **b** Representative images of p21 (red) in cholangiocytes (green) in mice treated with vehicle, 10 or 20 mg/kg of LY2157299. **c** Percentage of p21-positive cholangiocytes diminishes with the administration of LY2157299. **d** Total percentage of p21 diminishes with LY2157299 administration. **e** Decrease of p27 total number of cells with LY2157299 administration. **f** Percentage of tdTom-positive cholangiocytes is not altered. Statistic analysis for **c–f** * denotes $p < 0.05$, ** denotes $p < 0.01$, *** denotes $p < 0.001$, **** denotes $p < 0.0001$ (Mean ± SEM). ANOVA, Sidak's multiple comparisons test ($N = 5$ per group). **g** After induction of senescence in cholangiocytes, LY2157299 was administered to the mice by oral gavage. Then DDC diet was administered to the mice for 1 week. Experimental groups for this experiment include: Senescence + Vehicle + DDC (Veh, $N = 5$) and Senescence + Inhibitor + DDC (Inh, $N = 5$). **h** Decrease in serum transaminase levels. **i** Increased levels of Ki67-positive cholangiocytes with the use of LY2157299. Far right, quantification. **j** Increased levels of total Ki67 per field with LY2157299. **k** Trend to decrease ($p = 0.0952$) of total p27 levels. **l** Left, decreased expression of p21-positive hepatocytes. Right, decreased expression of p16-positive hepatocytes. **m** Decreased collagen deposition with the use of LY2157299. **n** Trend to decrease ($p = 0.1508$) of total number of F4/80 macrophages per field. * denotes $p < 0.05$, ** $p < 0.01$ (Mean ± SEM). Student's *t*test. Scale bars = 50 μm

normalization was performed using the sliding parabola module. For each experiment identical thresholds were used in all images for assigning nuclei to a specific population.

For pixel analysis, ImageProPremier software was used to select regions of positivity and automatically analysed using a macro-instruction. Results are expressed as the mean percentage of positive pixels per field. PiSR analysis used an AxioScan Z.1 (Zeiss) to acquire tiled images at ×20 magnification. Images were then analysed using a standard colour threshold in Fiji Image J.

DAPI, Alexafluor 488, and 555 were detected using band paths of 415–480, 495–540 and 561–682 nm for 405, 488 543 nm lasers respectively. All scale bars denote 50 µm.

**Isolation of hepatocytes**. Mice were killed and their livers were perfused in situ with Liver Perfusion Medium (Gibco) and Liver Digest Medium (Gibco). The liver was then disrupted to yield a cell suspension, filtered through a 50 µm filter (BD Biosciences) and hepatocytes were pelleted by centrifugation at $135 \times g$ 1 min. Hepatocytes were then purified as previously described[43]. Briefly, cells were underlayered with 1.06, 1.08 and 1.12 g/ml Percoll (Sigma) in PBS. The 1.08 mg/ml was spiked with phenol red to aid visualization. Cells were spun at $750 \times g$ 20 min. Fraction between the 1.08 and 1.12 mg/ml Percoll layers was harvested, purified again and transferred to Trizol (Life Technologies). Routinely, cells with hepatocyte morphology and expression of CYP2D6 (marker of hepatocellular differentiation) were obtained at over 99% purity[29].

**Isolation of bile ducts**. Bile duct isolation was performed as previously described[44]. Briefly, livers were perfused with saline, dissected down to approximately 2–4 mm³ portions and covered with a mix of Collagenase-Dispase in 2%FCS Advanced DMEMF/12 (Thermo Fisher). This mix was incubated at 37 °C 1 h and individual ductular structures were manually picked up, cleaned with 5%FCS DMEM (Thermo Fisher) and transferred to Trizol.

**Isolation of BMDM and co-culture with senescent biliary cells**. BMDM were isolated as previously described[45], and stained with 647-CellMask Deep Red Plasma membrane stain (ThermoFisher) following the manufacturer's instructions. Biliary cells were treated with CM-SASP for 1 week and washed with PBS1x five times. Number of cells were counted and the same ratio of pre-stained 647-BMDM was added. Both populations were cultured for 72 h.

**Immunoblotting**. Ductular structures were manually picked up, cleaned with 5% FCS DMEM and protein extracted with RIPA (Sigma) according to the manufacturer's instructions. Protein content was quantified using a BSA standard and Pierce reagent before being resolved using SDS/PAGE and blotted onto nitrocellulose membrane. Membranes were blocked with 5% BSA on T-PBS during 1 h at room temperature. Membranes were incubated at 4 °C overnight with primary antibody (mouse anti-β actin (Sigma), rabbit anti-p-Smad2 (Ser465/567)/Smad3 (Ser 423/4225) (Cell Signalling) or rabbit anti-NF-kB p65 (Cell Signalling)). Appropriate HRP-labelled secondary antibodies were used (Dako or CST) and signal detected with ECL reagent (Amersham). In order to use the same membrane with different antibodies, stripping was performed with mild stripping buffer during 10 min at room temperature. Membrane was blocked again with 5% BSA on T-PBS during 1 h and re-incubated with the desired antibody overnight. All uncropped western blots can be found in Supplementary Figure 14.

**RNA isolation followed by RT-qPCR and gene expression analysis**. RNA was isolated from bile ducts and hepatocytes using Trizol (Life Technologies) and purified using RNease RNA extraction kit (Qiagen) according to the manufacturer's instructions. RNA concentration and quality was assessed using a Nanodrop spectrophotometer (Thermo-Fisher). Reverse transcription was performed using QuantiTect Reverse transcription kit (Qiagen). Real time-QPCR was performed on a LightCycler 480 II (Roche) with commercial primers from Qiagen's QuantiTect (stated in Supplementary Table 4). Each gene expression was assessed with its own standard curve and normalized using Peptidylprolyl Isomerase A (PPIA) as housekeeping gene. Samples were run in triplicate.

**Statistical analysis**. A minimum of four mice were included per experimental group. For cell culture experiments, a minimum of four independent biological replicates were performed.

Prism software version 5.0a (GraphPad Software, Inc) was used for all statistical analyses. Normal distribution of data was determined using D'Agostino and Pearson omnibus normality test with Welch's correction if variances differed (F test). For parametric data, data significance was analysed using a two-tailed unpaired Student's $t$-test. Non-parametric data was analysed using Mann−Whitney test. In cases where more than two groups were being compared, then a one-way ANOVA (with Bonferroni correction) was used. Survival curves are calculated using Log-Rank comparison (Mantel−Cox test) and Gehan−Breslow−Wilcoxon test.

Statistical significance was assumed at $p < 0.05$. Data is presented as mean ± standard error of the mean (SEM). N refers to biological replicates.

**Data availability**. The authors declare that all data supporting the findings of this study are available within the article and its supplementary information files or from the corresponding author upon reasonable request. No data sets were generated or analysed during the current study.

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

## Acknowledgements

The authors would like to thank Prof. Lozano (Department of Cancer Genetics, University of Texas) for Mdm2flox/flox mice; and F. Lemaigre and I. Leclercq for OPNi-CreERT2R26RYFP mice. S.F.-G. was funded by Principal's Career Development Ph.D. Scholarships for Ph.D. students and via the MRC Centre for Regenerative Medicine Centre grant. This work was supported by the UK Medical Research MRC (MR/K017047/1), the Computational and Chemical Biology of Stem Cell Niche (MR/L012766/1) and the UK Regenerative Medicine Platform (MR/K026666/1).

## Author contributions

Conceptualization and design (S.F.-G., W.-Y.L., L.B., J.-C.A., T.G.B., S.J.F., L.C.). Data generation (S.F.-G., W.-Y.L., A.R., T.Y.M., B.D., T.J.K., N.T., P.J.S.L.). Data analysis and interpretation (S.F.-G., W.-Y.L., A.R., J.-C.A., E.O.D., S.J.F.). Manuscript preparation (S.F.-G., W.-Y.L., B.D., S.J.F.). Review and editing (S.F.-G., W.-Y.L., B.D., S.J.F.). Funding acquisition (S.J.F.).

## Additional information

**Competing interests:** The authors declare no competing interests.

