## [Peer Review File · Nature Communications]

Reviewers' Comments:

Reviewer #1:

Sofia Ferriera-Gonzalez et al., present a well-written manuscript which introduces a novel mouse model of biliary epithelial cell senescence. This model system will be a useful tool in determining the functional role of biliary epithelial cell senescence in the pathogenesis of the cholangiopathies, including Primary Sclerosing Cholangitis (PSC) and Primary Biliary Cholangitis (PBC).

Their model system utilizes an inducible Cre-Lox system to knockout MDM2, a negative regulator of p53, selectively in cholangiocytes. This system successfully induced senescence in the biliary epithelia and the surrounding parenchyma through SASP (TGFbeta). They showed that senescent cholangiocytes could exaggerate liver disease observed in a mouse model of PSC (DDC-diet). They also showed that senescent cholangiocytes negatively impacted the regenerative capacity of the liver after partial hepatectomy. While the authors present a strong manuscript, the following should be addressed prior to publication.

The weaknesses of this manuscript are

- (i) While their inducible Cre-Lox system successfully induced cellular senescence in the liver, the authors did not examine other K19 positive epithelia. K19 is expressed in various epithelia, including those in the intestine and kidney. Therefore, at a minimum, the authors need to examine these tissues for senescence and fibrosis.
- (ii) The authors neglected to look at the p16 signaling pathway, a major driver of cellular senescence. It is possible that p16 is not involved in driving biliary senescence in their model. However, the authors need to examine and discuss it. Are there p16 positive cholangiocytes? Are there any p16 positive cells in the parenchyma? Do isolated bile ducts overexpress p16?
- (iii) The authors extended their mice cohort for 90 days. They showed that senescent cholangiocytes remained viable. However, they did not examine liver disease progression. It would be predicted that these senescent cholangiocytes continue to produce SASP, which negatively impacts liver function. The authors need to check the biochemistry of the liver (ALP, ALT) and fibrosis.

Minor comments:

In figure 2:

There is a lack of consistency in labels. Some IF panels have the precise date after induction, while others don't have any label. This made it challenging to interpret results.

In figure 3:

The pSmad2/3 is not convincing. It looks like only one mouse has an increase (#4). Additionally, an increase in pSmad2/3 is evident in some of the Cre- +TM. The authors need to repeat or elaborate on this finding.

In figure 4:

The authors should add a WT group and/or a vehicle treated group to the figure panels. DDC treatment alone will increase serum biochemistries compared to untreated or vehicle treated. It would be beneficial to know how DDC, compared to untreated or vehicle control, alters Ki67 and p21 expression.

Other remarks:

The authors should carefully review the references. The citations in the body of the text have a few mistakes. For example, page number 28, reference number 29 is incorrect.

Reviewer #2:

Remarks to the Author:

This is a well written and illustrated paper that shows that biliary cell senescence can be a driving force for biliary damage and that senescence can also exacerbate other biliary injuries. The paper combines both in vitro and in vivo descriptive and functional data to show a pivotal role for TGFβ in the transmission of a detrimental senescent phenotype to neighbouring parenchymal (hepatocyte) cells, and how this pathway may be targeted for disease alleviation. I have a few points that are dealt with in order

1. Figure 1 mentions 'Explants'. As far as I can see these are resections or biopsy specimens. An Explant usually refers to a tissue removed from the body and cultured in a medium.
 2. Page 7 mentions 'a large polymorphonuclear infiltrate identified as F4/80-positive macrophages'. Polymorphonuclear refers to the likes of neutrophils and eosinophils., replace with 'mononuclear infiltrate'
 3. Please explain some seemingly inconsistencies between Figures. In Fig. 2c and 3a there are generally few p53+ cholangiocyte nuclei, whereas much more abundant in 3e. Fig. 3e, why are the p53+ bile ducts (green) not expressing the likes of 53BP1 and DCR2 as shown in 2d? There seems to be a huge discrepancy in the numbers between those hepatocytes expressing p27 (very few in 3c) and those expressing 53BP1, H2A.X and DCR2 (most hepatocytes in 3e).
 4. The labeling of many of the quantitative data graphs is ambiguous. For example 4e, j, and l, 5c and 5e, 7d, e, k and l all have units expressed as e.g. % F4/80/field or %Ki67/field. As far as I can tell these are %s calculated from counting cells in 30 fields at 200x mag'n (Methods. 6. Microscopy and cell counting). Therefore the word 'field' is redundant and will confuse readers in to thinking this has something to do with counting per HPF.
 5. Page 18 bottom line "through a TGFβ-dependent SASP mechanism' The authors are being premature in this statement. The functional data proving this are introduced after this statement.
- Malcolm Alison

Reviewer #3:

Remarks to the Author:

Relationship between cellular senescence and cholangiopathy has been known and there are several recent papers describing the mechanism. It has been also shown that SASP including TGFβ derived from bile ducts in senescence affect surrounding cells. Thus, conceptual advance by this paper may be limited. However, previous reports were mostly based on in vitro experiments. In this paper, the authors used their own in vivo model to demonstrate causal relationship. The model can be used to manipulate senescence in vivo and would be useful.

It is not clear why the authors focused on TGF-β among various SASP and whether TGF-β derived from bile ducts directly affects bile ducts and hepatocytes.

The authors used floxed K19-Mdm2 and liver specific tdTom mice, and show expression of cellular senescence markers. However, it is not clear whether cell cycle arrest was irreversible or transient. Additional evidence would be necessary.

In this mouse model, not only senescence but also apoptosis was induced (Sup Fig. 2d) . Thus, it is not clear whether aggravation of liver injury was due to senescence, apoptosis, or both.

Recombination by K19-CreERT is not specific to bile ducts because K19 is expressed in mesothelial cells. Senescence of mesothelial cells needs to be examined. As mesothelial cells express various growth factors for hepatocytes, regeneration after PHx could be affected by mesothelial cells.

Fig. 1

There appear to be many hepatocytes with DCR2 and H2a. It is not clear whether they were background or real signals. Isotype data should be presented.

Fig. 2c. The authors state that Ki67+ cells in p53+ cells were 0.1%. However, the data seem to show more Ki67+ cells. Are they p53 negative? More clear data should be provided. By staining K19, Ki67, and p53, Ki67 positive fraction in p53+ and p53- cells should be quantified.

Fig 2d. The authors state in text that cellular senescence was induced in cholangiocytes. However, the cell should be hepatocyte.

Supplementary Fig. 2a

The authors state that bile duct senescence induced ductular reaction but tdTom+ cholangiocyte senescence was not altered (p7). As ductular reaction accompanies cholangiocyte proliferation, the relative tdTom+ fraction is expected to decrease. But it was not decreased, suggesting that tdTom+ cholangiocytes were not irreversibly arrested.

Supplementary Fig. 4e and 4f and Fig. 3a

Based on p21+/tdTom- or YFP- hepatocytes and p21+/p53- cholangiocytes, the authors argue senescence in a paracrine manner. However, there are many other possibilities to be considered. For example, without recombination of tdTom or YFP, cholangiocytes with recombination of the Mdm2 locus. Such cells could be converted to hepatocytes. Presence of cholangiocytes that were induced to express p21 by loss of Mdm2 and p53 expression. Expression of Mdm2 should be examined in the senescent cells as a result of paracrine manner.

Fig. 3c and 3d

Was p27 detected in bile ducts? According to the conclusion by the authors, p53+ primary senescent cholangiocytes do not express p27, but only p53- secondary senescent cholangiocytes express p27.

Fig. 3h

Activation of TGF-beta signaling in hepatocytes should be confirmed.

Fig. 4

The mouse line shows spontaneous ductular reaction without DDC (Sup Fig. 2a) . Was DDC-induced ductular reaction enhanced or reduce?

The statement (line 113) "We also found increased numbers of α SMA-positive cells surrounding the senescent 114 cholangiocytes (Fig. 2g and Supplementary Fig. 3)" is not appropriate, because the authors measured fluorescence intensity of α SMA surrounding the bile duct. For this claim, the cell number should be counted.

Fig. 6c and 6d

Specificity of YFP+ biliary cells on the effect of inhibition of TGF-beta signaling needs to be shown.

Fig. 7f

Effect of TGF-beta inhibitor. The authors state that the number of primary senescent cholangiocytes (tdTom-positive) remain unaltered, proving that primary senescence is not affected by the

administration of LY2157299] (page 25, line 376) . However, tdTom+ cells were created by Cre-mediated recombination but was independent on senescence induction.

Page 11, line 182

[Other members of the TGF-beta family, such as Tgfr1 and Tgfr2]
Tgfr1 and Tgfr2 are receptors, but not TGF-beta family members.

In the in vitro system, macrophages removed senescent cells, whereas in vivo results (Fig. S2a) showed that tdTomato+ cells were not reduced. This discrepancy needs explanation.

Reviewer #4:

Remarks to the Author:

The authors generated a model of biliary senescence by conditionally knocking out Mdm2 in bile ducts. Senescent cholangiocytes induced profound alterations in the microenvironment with recruitment of myofibroblasts and macrophages causing collagen deposition, TGFbeta production and induction of senescence in surrounding cells. This new model supports the concept of paracrine transmission of senescence. The authors identified TGFbeta as a major mediator of this response.

The in vitro and in vivo experiments are very convincing and provide a rationale for targeting TGFbeta to treat biliary disease and accompanying liver damage.

Comments:

1. The TGFbeta type I receptor (ALK5) inhibitor used in the study (LY2157299) is not specific as it will also block signalling through ALK4/ALK7 - this is an important point to include.
2. In fact, it would be great for the field if the TGFbeta ligand(s) contributing to the paracrine transmission of senescence could be identified. The authors show that TGFbeta1 mRNA levels are elevated in their model, but what about TGFb2 and TGFb3, which utilise the same signalling pathway? Or activin A and activin B, which utilise ALK4/ALK7 and have previously been implicated in cholangiocyte development and liver regeneration. Showing which of these ligands contributes to senescence would be a major step forward, as it would enable the utilisation of more specific inhibitors.
3. If other TGFbeta ligands are upregulated, then their importance could be examined in the in vitro model.
4. In Fig. 7B, it is difficult to see the p21+ cells - can these be expanded?
5. Fig 6A - Should read "Quantify" and "Quantification"

Dear Reviewers- we are grateful for the careful review of our work. We are pleased with the enthusiasm shown for this manuscript and the constructive feedback. We have addressed the points below.

REVIEWER 1.

Sofia Ferreira-Gonzalez et al., present a well-written manuscript which introduces a novel mouse model of biliary epithelial cell senescence. This model system will be a useful tool in determining the functional role of biliary epithelial cell senescence in the pathogenesis of the cholangiopathies, including Primary Sclerosing Cholangitis (PSC) and Primary Biliary Cholangitis (PBC).

Their model system utilizes an inducible Cre-Lox system to knockout MDM2, a negative regulator of p53, selectively in cholangiocytes. This system successfully induced senescence in the biliary epithelia and the surrounding parenchyma through SASP (TGFbeta). They showed that senescent cholangiocytes could exaggerate liver disease observed in a mouse model of PSC (DDC-diet). They also showed that senescent cholangiocytes negatively impacted the regenerative capacity of the liver after partial hepatectomy. While the authors present a strong manuscript, the following should be addressed prior to publication.

The weaknesses of this manuscript are:

(i) While their inducible Cre-Lox system successfully induced cellular senescence in the liver, the authors did not examine other K19 positive epithelia. K19 is expressed in various epithelia, including those in the intestine and kidney. Therefore, at a minimum, the authors need to examine these tissues for senescence and fibrosis. Thank you for pointing this out. We have analyzed other K19-positive tissue such as Kidney and Gut (see **Supplementary Figure 6**). As expected, both of them present p21-senescent epithelial cells. Fibrotic deposition is exacerbated as time passes (compare Picrosirius red accumulation at day 2 and day 90).

From left to right: Left: Cre-negative mice at day 2 after tamoxifen administration. Center: Cre-positive mice at day 2 after tamoxifen administration Right: Cre-positive mice at day 90 after tamoxifen administration. **(a)** p16 immunohistochemistry in Cre- +TM at day 2 shows low expression of p16 in kidney and gut epithelia. After induction, p16 is expressed in both epithelia at day 2 and day 90. **(b)** Collagen accumulation assessed by PiSR staining shows increased deposition over the course of time in gut and kidney epithelia. Scale bars = 50µm.

(ii) The authors neglected to look at the p16 signaling pathway, a major driver of cellular senescence. It is possible that p16 is not involved in driving biliary senescence in their model. However, the authors need to examine and discuss it. Are there p16 positive cholangiocytes? Are there any p16 positive cells in the parenchyma? Do isolated bile ducts overexpress p16?

Thank you for this observation. We have included immunohistochemistry for p16 in our model (**Supplementary Figure 4a**). After induction of senescence, p16 is accumulated in cholangiocytes. Furthermore, p16-positive hepatocytes are found in the parenchyma in close proximity to the p16-positive cholangiocytes at day 21 and day 90 post-induction.

Legend: p16-immunohistochemistry shows accumulation of p16 positive cells in cholangiocytes in Cre+ +TM mice at day 2 after induction. Some hepatocytes express p16 as well (red arrows, areas surrounded in red). From left to right: Left: Cre-negative mice at day 2 after tamoxifen administration. Center: Cre-positive mice at day 2 after tamoxifen administration. Right: Cre-positive mice at day 90 after tamoxifen administration. N=4-6 mice per group.

Supporting this result, we found that the expression of *cdkn2a* is significantly increased in isolated bile ducts ($p=0.0006$), **Supplementary Figure 9b**).

Legend: Isolated bile ducts express *cdkn2a* in induced mice (Cre+ +TM) when compared with un-induced mice (Cre- +TM) at day 2 after tamoxifen administration.

Furthermore, use of TGFb inhibitor (LY2157299) in the setting of Senescence+DDC in our K19MDM2 model results in a significant decrease of total p16 expression (**Figure 7.I**)

Legend: Total percentage of p16-positive cells diminishes with the administration of LY2157299 in the K19-MDM2 model after tamoxifen-induction and 1 week of DDC diet. N=5 mice per group.

(iii) The authors extended their mice cohort for 90 days. They showed that senescent cholangiocytes remained viable. However, they did not examine liver disease progression. It would be predicted that these senescent cholangiocytes continue to produce SASP, which negatively impacts liver function. The authors need to check the biochemistry of the liver (ALP, ALT) and fibrosis.

We have analyzed total percentage of PicroSirius red staining in induced mice (Cre+ +TM) at day 2, day 21 and day 90 after induction. Results show a significant increase of fibrotic response as time passes (see **Supplementary Figure 5d**).

Legend: Quantification of collagen deposition assessed by PiSR staining over the course of 90 days after the induction of the model. * denotes $p < 0.05$, ** denotes $p < 0.01$, (Mean \pm SEM). ANOVA, Sidak's multiple comparisons test. N=4-6 per group.

Furthermore, the significant increase of transaminases at day 90 in comparison with day 2 after induction, suggest that liver function is further impaired as time passes (see **Supplementary Figure 3d**).

Legend: Liver transaminases in healthy control, Cre- +TM at day 2 and Cre+ +TM at day 2, 21, 60 and 90 after tamoxifen administration. Significant increase of ALT, AST and Bilirubin at day 90 compared with day 2 post-induction. The rest of transaminases (ALB, ALP and Cholesterol showed a trend to increase). N=4-6 per group.

We have also now included ductopenic index over the course of time in **Supplementary Fig. 3e**. There are no significant changes in the number of bile ducts per portal tract over the course of 90 days.

Legend: Ductopenic index shows no significant alterations in the course of 90 days. The blue line depicts the baseline level of ductopenia. ANOVA, Sidak's multiple comparisons test. (N=4-6 per group).

Minor comments:

In figure 2: There is a lack of consistency in labels. Some IF panels have the precise date after induction, while others don't have any label. This made it challenging to interpret results.

Thank you for pointing this out. We have now included the date after induction in all the panels of this figure, which will hopefully help to understand these results.

In figure 3: The pSmad2/3 is not convincing. It looks like only one mouse has an increase (#4). Additionally, an increase in pSmad2/3 is evident in some of the Cre- +TM. The authors need to repeat or elaborate on this finding.

Thank you for pointing this out. We have changed the quantification data in **Figure 3h** from a column to a scatter dot-plot, to help to identify the levels of NFkB and pSmad2/3 in the isolated bile ducts per mouse (normalized to bActin).

(i). qRT-PCR of *Nfkb* in hepatocytes isolated from Cre- and Cre+ K19-Mdm2^{fllox/fllox}tdTom^{LSL} mice at day 2 after last tamoxifen administration (N=5-6 per group).

From this data it become obvious that:

1. Some Cre- +TM isolated bile ducts express basal levels of pSmad2/3 and NFkB
2. There is a high variability in the Cre+ +TM group, which could indicate that not all bile ducts respond to tamoxifen in the same manner. This variability also correlates with the variability of Cre-dependent recombination (assessed by TdTom response per bile duct) as seen in **Supplementary Figure 7c**. Recombination rates in our model range from 30 to 68%, which could explain the variability in the senescent response.

We acknowledge that there some animals present pSmad2/3 and NFkB activation in the Cre- +TM group. This result is consistent with the gene analysis data from **Figure 3f, g**, where the baseline levels in the Cre- +TM control show activation of *Tgfb* and pro-inflammatory cytokines such as *Il6*.

Furthermore, the newly added RNA Scope for *Tgfb1* in Cre- +TM mice at day 2 after tamoxifen induction shows some *Tgfb1*-positive cholangiocytes and hepatocytes (**Supplementary Figure 9e**). This data supports this activation of pSmad2/3 signaling pathway in Cre- mice (see image below).

Legend: ISH by RNAscope for *Tgfb1* in Cre- +TM and Cre+ +TM at day 2 (N=3 per group). For the Cre- +TM, the selected areas show the absence of *Tgfb1* in cholangiocytes. For the Cre+ +TM, selected areas are magnifications of cholangiocytes (bottom left) and hepatocytes (bottom right) that express *Tgfb1*. Notice the presence of some *Tgfb1*-positive macrophages in the Cre+ +TM group. Right *PPIB* positive control (N=1). Scale bars = 50 μ m

We have noticed that tamoxifen administration causes some degree of biliary damage (see the levels of bilirubin in the image below), which could potentially explain the activation of these signaling pathways in our model. In fact, some studies already pointed out a direct relationship between the use of tamoxifen and hepatotoxicity¹⁻⁴

Legend: Bilirubin levels in C57BL6J Healthy mice and mice administered with 1, 2 or 3 intraperitoneal doses of tamoxifen in alternate days increases biliary injury (we consider pathological levels of bilirubin above 10 μ mol/l, See Supplementary Figure 3d). * denotes $p < 0.05$, ** denotes $p < 0.01$, (Mean \pm SEM). ANOVA, Sidak's multiple comparisons test.

1. Floren LC et al. Tamoxifen in liver disease: potential exacerbation of hepatic dysfunction. *Ann Oncol.* 1998 9(10): 1123-6.
2. Gao FF et al. Tamoxifen induces hepatotoxicity and changes to hepatocyte morphology at the early stage of endocrinotherapy in mice. *Biomed Rep.* 2016. 4(1):102-106.
3. De Conti A et al. Genotoxic, epigenetic and transcriptomic effects of tamoxifen in mouse liver. *Toxicology.* 2014. 325: 12-20.
4. Cole, LK; Jacobs, EL and Vance, DE. Tamoxifen induces triacylglycerol accumulation in the mouse liver by activation of fatty acid synthesis. *Hepatology.* 2010 52(4): 1258-65.

In figure 4: The authors should add a WT group and/or a vehicle treated group to the figure panels. DDC treatment alone will increase serum biochemistries compared to untreated or vehicle treated. It would be beneficial to know how DDC, compared to untreated or vehicle control, alters Ki67 or p21 expression.

We have now included in **Supplementary Figure 10a, b, c, d** and **e** a comparison between:

- **C57BL6 Healthy**: Healthy WT-C57BL6/J group (the same background as the our K19-MDM2-TdTom model).
- **C57BL6 DDC**: 1 week DDC in WT-C57BL6/J group.
- **K19Mdm2 Oil + DDC**: 1 week DDC in the uninduced K19-MDM2-TdTom model (administered with the vehicle, oil). This is the same group of Figure 4).

As expected there is an increase of Ki67 and p21 with DDC diet when compared with the healthy group. This is also the case for transaminases' levels. We haven't found significant differences between DDC and Vehicle+DDC groups (see below).

Legend: (a) Liver transaminases (alanine transaminase, aspartate aminotransferase, cholesterol and bilirubin). (b) Total percentage of proliferating (Ki67-positive) K19-positive cells. (c) Total percentage of senescent (p21-positive) K19-positive cells. (d) Total percentage of proliferating (Ki67-positive) cells. (e) Total percentage of senescent (p21-positive) HNF4a-positive hepatocytes.

Other remarks:

The authors should carefully review the references. The citations in the body of the text have a few mistakes. For example, page number 28, reference number 29 is incorrect.

Thank you very much for this careful observation (we skipped one number in the text). We have now fixed this and, for example, reference 29 is now Tabibian et al., 2016 (which is actually a seminal paper in senescence and biliary disease).

REVIEWER 2.

This is a well written and illustrated paper that shows that biliary cell senescence can be a driving force for biliary damage and that senescence can also exacerbate other biliary injuries. The paper combines both in vitro and in vivo descriptive and functional data to show a pivotal role for TGFbeta in the transmission of a detrimental senescent phenotype to neighbouring parenchymal (hepatocyte) cells, and how this pathway may be targeted for disease alleviation. I have a few points that are dealt with in order

1. Figure 1 mentions 'Explants'. As far as I can see these are resections or biopsy specimens. An Explant usually refers to a tissue removed from the body and cultured in a medium.

Thanks for pointing this out. In the methods sections we wrote biopsies. We have now changed to biopsies in the text as well.

2. Page 7 mentions 'a large polymorphonuclear infiltrate identified as F4/80-positive macrophages' Polymorphonuclear' refers to the likes of neutrophils and eosinophils., replace with 'mononuclear infiltrate' We do completely agree. We have changed to mononuclear infiltrate.

3. Please explain some seemingly inconsistencies between Figures. In Fig. 2c and 3a there are generally few p53+ cholangiocyte nuclei, whereas much more abundant in 3e.

For the manuscript, we tried to include representative images of each immunofluorescence, which sometimes may not match the quantitative data provided.

In the K19-Mdm2 model we have found a high variability in the Cre-dependent recombination response (ranging from 31 to 68% of recombination, as seen in **Supplementary Figure 7.c, right**). This could potentially account for the apparent differences in the number of p53+ cholangiocytes in the different images.

Legend: Primary senescence in cholangiocytes is similar in both models. Images show YFP-positive cholangiocytes (green) and tdTom-positive (red) K19-positive cholangiocytes (grey). Upper corner of the image shows a digital magnification of representative areas. Far right, quantification of the number of YFP per K19-positive cholangiocytes (OPN-Mdm2^{flox/flox}YFP model) vs tdTom per K19-positive cholangiocytes (K19-Mdm2^{flox/flox}tdTom^{LSL}). Images show uninduced (Cre+ -TM) versus induced (Cre+ +TM) mice at day 2 after last tamoxifen injection.

Fig. 3e, why are the p53+ bile ducts (green) not expressing the likes of 53BP1 and DCR2 as shown in 2d?

We apologize for the quality of **Figure 3e**, the exposure of p53 in the green channel covers the expression of 53BP1, H2A.X and DCR2 (red). In the original images, p53+ bile ducts express 53BP1, DCR2 and H2A.X. We have included in **Supplementary Figure 8a** single stainings of **Figure 3e** to prove the expression of these markers in the bile ducts. For p53+DCR2 double immunofluorescence, as DCR2 expression is cytoplasmatic, we have included an area of extrahepatic bile ducts (where cholangiocytes have a higher cytoplasm/nuclei ratio) so the staining is more clear.

a

Legend: Single channels for Figure 3e (Fig 3. Paracrine senescence in the model is TGF β -dependent). Double immunostaining for p53 (green) and several senescent markers in Cre+ +TM mice at day 2 after induction (N=6 per group). From left to right: 53BP1, H2A.X and DCR2 (red). From up-down: Merge, p53, senescent marker, DAPI. Scale bars = 50 μ m.

There seems to be a huge discrepancy in the numbers between those hepatocytes expressing p27 (very few in 3c) and those expressing 53BP1, H2A.X and DCR2 (most hepatocytes in 3e).

To explain this discrepancy, we have quantified H2A.X, DCR2, 53BP1, p21 and p27 in hepatocytes (included in **Supplementary Figure 8b**).

While there is a high variability between animals, these data suggest that the paracrine response in the parenchyma is similar in terms of expression of p27, p21 or the different senescent markers (53BP1, DCR2 and H2A.X). The images used in the manuscript are intended to be representative images. However, the animals depicted are selected randomly for each image, increasing the variability in between figures.

4. The labeling of many of the quantitative data graphs is ambiguous. For example, 4e, j, and l, 5c and 5e, 7d, e, k and l all have units expressed as e.g. % F4/80/field or %Ki67/field. As far as I can tell these are %s calculated from counting cells in 30 fields at 200x magnification (Methods. 6. Microscopy and cell counting). Therefore, the word 'field' is redundant and will confuse readers in to thinking this has something to do with counting per HPF.

We do agree. Quantifications are made by counting cells in 30 fields per slide, and adding "field" to the graph axis only adds more confusion. We have just indicated the percentage and eliminate the word field from the Y axis of the figures.

5. Page 18 bottom line "through a TGFb-dependent SASP mechanism' The authors are being premature in this statement. The functional data proving this are introduced after this statement.

We do agree, the role of TGFb in the system is disclosed in the in vitro experiment in the next figure. We have changed to: "These results suggest that cholangiocyte senescence impairs the regenerative response of the liver parenchyma. This could potentially account for the loss of hepatocyte function seen in human PSC/PBC patients."

In the next section we introduce the potential relevance of TGFb in the model with:

"To test whether transmission of senescence from cholangiocytes to the surrounding hepatocytes is TGFb-dependent, we generated an *in vitro* model based..."

REVIEWER 3.

Relationship between cellular senescence and cholangiopathy has been known and there are several recent papers describing the mechanism. It has been also shown that SASP including TGFb derived from bile ducts in senescence affect surrounding cells. Thus, conceptual advance by this paper may be limited. However, previous reports were mostly based on in vitro experiments. In this paper, the authors used their own in vivo model to demonstrate causal relationship. The model can be used to manipulate senescence in vivo and would be useful.

1. It is not clear why the authors focused on TGF-b among various SASP and whether TGF-b derived from bile ducts directly affects bile ducts and hepatocytes.

We initially focused into TGFb signaling as, in our model, *TGFb1* presented the highest significant variance from all the assessed SASP's factors, i.e. *TGFb1* ($p=0.0007$) vs *Nfkb* ($p=0.0132$), *Il1a* ($p=0.0160$) or *Il6* ($p=0.1708$).

Expression of p27, a direct target of TGFb signaling pathway¹ in hepatocytes (see **Figure 3c, d** and **Supplementary Figure 9b**) suggest a direct functional role for TGFb in the expansion of paracrine senescence response.

A functional proof of the role of TGFb in our model arises from the use of LY2157299 which directly blocks TGFb signaling pathway (*in vitro* in **Figure 6** and *in vivo* in **Figure 7**). The use of this small molecule in our K19-MDM2 model (**Figure 7**) decreased the levels of paracrine senescence in the parenchyma and proved the role of TGFb in the transmission of senescence to the liver parenchyma.

We do acknowledge that TGFb is just one single factor in a myriad of SASP components. At the time of these experiments (and taking into consideration the importance of TGFb in the transmission of the SASP, as shown by Acosta and colleagues²) we considered the use of TGFb inhibitors to partially stop the spreading of paracrine senescence. While this approach partially blocks the effects of senescence, it improves features of biliary injury in our model, such as fibrosis. Furthermore, it proves the importance of TGFb in senescence-related biliary injury.

1. Muñoz-Espín D, Serrano M. Cellular senescence: from physiology to pathology. *Nat Rev Mol Cell Biol.* 2014; 15(7): 482-496.
2. Acosta, JC. et al. A complex secretory program orchestrated by the inflammasome controls paracrine senescence. *Nat Cell Biology.* 2013; 15(8): 978-990.

2. The authors used floxed K19-Mdm2 and liver specific tdTom mice, and show expression of cellular senescence markers. However, it is not clear whether cell cycle arrest was irreversible or transient. Additional evidence would be necessary.

We have now included a quantification of proliferating (Ki67-positive) recombined (TdTom-positive) cells in **Supplementary Figure 2e**. This figure shows that percentage of recombined (senescent cholangiocytes) is very low and does not vary with the course of time (final time point 90 days), suggesting that cell cycle arrest is irreversible.

Legend: Quantification of proliferation (assessed by Ki67) in TdTom-positive cholangiocytes shows no significant differences over the course of 90 days. ANOVA, Sidak's multiple comparisons test. N=4-6 per group.

3. In this mouse model, not only senescence but also apoptosis was induced (Sup Fig. 2d) . Thus, it is not clear whether aggravation of liver injury was due to senescence, apoptosis, or both.

We have calculated the number of TUNEL-positive cholangiocytes in our model up to day 90 post-induction (see **Supplementary Figure 4d**). Number of TUNEL-positive cells appear to increase at day 21 and 90 compared

with day 2, suggesting a progressive course for apoptosis. This late appearance of TUNEL-positive cells might indicate that apoptosis is a secondary consequence and/or an off-target effect of the induction of senescence

in our model.

Legend: TUNEL staining (red) shows an increase in apoptotic cholangiocyte nuclei in the Cre+ mice 21 days after induction. N= 4-6 per group. **(d)** Quantification of TUNEL-positive cholangiocytes over the course of 90 days after induction of the model. The image includes isotype control and DNaseI-positive control. ** denotes $p < 0.01$, (Mean \pm SEM). ANOVA, Sidak's multiple comparisons test. N= 4-6 per group. Scale bars = 50 μ m.

We do agree that, in our model, apoptosis can potentially amplify the effect of senescence to exacerbate biliary injury as has been previously postulated¹. We have added this data and comment to the paper.

1. Sasaki and Nakanuma. Biliary epithelial apoptosis, autophagy and senescence in primary biliary cirrhosis. *Hepat Res Treat.* 2010: 205128.

4. Recombination by K19-CreERT is not specific to bile ducts because K19 is expressed in mesothelial cells. Senescence of mesothelial cells needs to be examined. As mesothelial cells express various growth factors for hepatocytes, regeneration after PHx could be affected by mesothelial cells.

To evaluate the role of senescence in mesothelial cells in our model, we assessed the expression of a well known mesothelial marker: Wilms Tumor-1 (WT1) (Li, Wang and Asahina, 2013)¹.

In our model, WT1 is not expressed after induction of senescence (**Supplementary Fig. 6c**). Similarly, assessment at long term revealed no WT1 expression.

Legend: Immunohistochemistry for Wilms Tumor 1 (WT1) shows no involvement of mesenchymal cells in our model. This image includes isotype control and a positive control (Glisson Capsule of K19-Mdm2^{flox/flox}tdTom^{LSL} mice, connective tissue septa). Scale bars = 50µm.

In 70% partial hepatectomy (PH, with and without senescence) in which active proliferation of parenchymal cells occurs, we did not observe WT1 expression (**Supplementary Fig. 11d**). This result backups previous data presented by *Perugorria and colleagues in 2009*, in which WT1 is not up-regulated in the context of PH².

Legend: Immunohistochemistry for Wilms Tumor 1 (WT1) shows no involvement of mesenchymal cells in our model after partial hepatectomy. This image includes isotype control and a positive control (Glisson Capsule of K19-Mdm2^{flox/flox}tdTom^{LSL} mice, connective tissue septa). Scale bars = 50µm

Altogether, these data suggest that mesothelial cells do not play an active role in our model and that regeneration after PH is, in this case, not affected by mesothelial cells.

1. Li, Y., Wang, K., Asahina, K. Mesothelial cells give rise to hepatic stellate cells and myofibroblasts via mesothelial-mesenchymal transition in liver injury (2013). PNAS 110(6): 2324-9.
2. Perugorria, MJ. et al. Wilms' Tumor 1 gene expression in hepatocellular carcinoma promotes cell dedifferentiation and resistance to chemotherapy (2009). Cancer Res 69(4):1358-67.

5. There appear to be many hepatocytes with DCR2 and H2a. It is not clear whether they were back ground or real signals. Isotype data should be presented.

Thank you for pointing this out. We have included isotype data for H2A.X as well in **Figure 1c, d**.

Legend: PSC and PBC show DCR2 expression (green) in cholangiocytes (red). Far right, quantification. **(d)** PSC and PBC show γ H2A.X expression (green) in cholangiocytes (red). Far right, quantification. ** denotes $p < 0.01$, *** denotes $p < 0.001$, **** denotes $p < 0.0001$ (Mean \pm SEM). ANOVA, Sidak's multiple comparisons test. Scale bars = 50 μ m.

We have also included images of the liver parenchyma in human PBC, PSC explants in **Supplementary Figure 1a, b**. This images indicate that H2A.X and DCR-positive hepatocytes are quite common in biliary disease.

Legend: (a) H2A.X (green) is present in K19-cholangiocytes (red) and surrounding hepatocytes in PSC and PBC. Right, same images without DAPI. **(b)** DCR2 (green) is present in K19-cholangiocytes (red) and surrounding hepatocytes in PSC and PBC. Right, same images without DAPI.

In a previous publication by Sasaki and colleagues¹, p21-positive hepatocytes can be observed in late stages of PBC (see the image on the right). Although they did not elaborate on this finding, the fact that in our batch of samples some hepatocytes present markers of senescence (such as p16, DCR2 and H2A.X) reinforce this finding.

This suggest that in late stages of PBC and PSC, the senescence footprint expands from cholangiocytes into the liver parenchyma, exacerbating biliary disease.

1.Sasaki et al. Frequent cellular senescence in small bile ducts in primary biliary cirrhosis: a possible role in bile duct loss. *J Pathol* (2005). 205(4):451-9.

This figure represents p21 staining in PBC human section and belongs to **Figure 2 Panel F** in *Sasaki et al. 2005*.

6. Fig. 2c. The authors state that Ki67+ cells in p53+ cells were 0.1%. However, the data seem to show more Ki67+ cells. Are they p53 negative? More clear data should be provided. By staining K19, Ki67, and p53, Ki67 positive fraction in p53+ and p53- cells should be quantified.

Yes, only 0.1% of p53-positive cells are Ki67-positive.

We have now included in **Supplementary Figure 2c** the percentage of proliferating cells per bile duct in the p53-positive and p53-negative fractions at Day 2 post-induction of the model.

This result show that p53-positive cholangiocytes have a very low proliferative rate (with an average of 0.1% Ki67+ p53+), whereas p53-negative cholangiocytes have a significant increased proliferate capacity.

Legend: Quantification of the total percentage of p53-positive and p53-negative cholangiocytes that proliferate in the Cre+ +TM mice at day 2. This result shows that p53-positive cells do not proliferate while p53-negative cells do. *** denotes $p < 0.001$, (Mean \pm SEM) Student's t-test, (N=6 per group).

As Mosteiro et al¹ and Ritschka et al² confirmed, senescent cells induce profound alterations in their environment that are key to induce tissue regeneration and cellular plasticity. This could be potentially the case in our model, in which, for example the cholangiocytes or the Hepatic Progenitor Cell compartment could be proliferating in response to this senescent cues. However, this is highly speculative and further experimentation would be required to test this hypothesis.

1. Mosteiro, L. et al. Tissue damage and senescence provide critical signals for cellular reprogramming in vivo. *Nat Science*. 2016; 354 (6315).
2. Ritschka, B. et al. The senescence-associated secretory phenotype induces cellular plasticity and tissue regeneration. *Genes Dev*. 2017. 31(2):172-183.

7. Fig 2d. The authors state in text that cellular senescence was induced in cholangiocytes. However, the cell should be hepatocyte.

We wanted to express that senescence is initially established in cholangiocytes. However, as Reviewer 3 pointed out, there are some hepatocytes in this figure that express markers of senescence. As shown in following figures, this might suggest the presence of paracrine-senescence mechanisms from cholangiocytes towards the liver parenchyma. To clarify this point, we have now changed the text to:

'Expression of other senescence markers like 53BP1, γ H2A.X and DCR2 (Fig. 2d), suggest the occurrence of cellular senescence in cholangiocytes. Furthermore, some hepatocytes also express 53BP1, γ H2A.X and DCR2 (Fig. 2d), suggesting the presence of paracrine activity from the senescent cholangiocytes towards the liver parenchyma.

8. Supplementary Fig. 2a

The authors state that bile duct senescence induced ductular reaction but tdTom+ cholangiocyte senescence was not altered (p7). As ductular reaction accompanies cholangiocyte proliferation, the relative tdTom+ fraction is expected to decrease. But it was not decreased, suggesting that tdTom+ cholangiocytes were not irreversibly arrested.

We have quantified again our data on K19-staining and TdTom staining and presented it in the following manner:

- Total Percentage of K19-positive cells (**Supplementary Figure 3, a**). This would hopefully give an idea of the increasing ductular reaction seen in the model (as shown by the significant increase at day 90 compared with day 2 post-tamoxifen administration).

- Total Percentage of TdTom-positive recombined cells (**Supplementary Figure 3, b**) This figure shows that at 90 days post tamoxifen-induction there is a non-significant trend to decrease in total number of senescent cells ($p=0.3823$), suggesting that our model establishes an irreversible cell cycle arrest.

* See answer 2.

Legend: (a) Quantification of the total percentage of TdTom-positive cells increases after induction of the model at day 2 but shows no significant differences over the course of 90 days. ANOVA, Sidak's multiple comparisons test. (b) Quantification of the total percentage of K19 significantly increases over the course of 90 days. ** denotes $p < 0.01$, (Mean \pm SEM). ANOVA, Sidak's multiple comparisons test. N=4-6 per group.

9. Supplementary Fig. 4e and 4f and Fig. 3a

Based on p21+/tdTom- or YFP- hepatocytes and p21+/p53- cholangiocytes, the authors argue senescence in a paracrine manner. However, there are many other possibilities to be considered. For example, without recombination of tdTom or YFP, cholangiocytes with recombination of the Mdm2 locus. Such cells could be converted to hepatocytes. Presence of cholangiocytes that were induced to express p21 by loss of Mdm2 and p53 expression. Expression of Mdm2 should be examined in the senescent cells as a result of paracrine manner.

We have performed in situ hybridization (ISH) for *Mdm2*. In control mice (Cre- +Tm), *Mdm2* shows a homogeneous pattern of expression in both cholangiocytes and hepatocytes (**Supplementary, Figure 2a**).

Legend: ISH by RNAscope for *Mdm2* in Cre- +TM (N=3) and Cre+ +TM (N=3) at day 2. For the Cre- +TM, the selected area shows the presence of *Mdm2* in cholangiocytes. For the Cre+ +TM, representative images of the three mice (N=1, 2 and 3) show the loss of *Mdm2* in cholangiocytes (in N=1) while hepatocytes still maintain *Mdm2* expression (as seen in the magnified area of N=2). Right PPiB positive control (N=1). Scale bars = 50µm

After induction (Cre+ +TM at day 2, 21 and 90), *Mdm2* expression in cholangiocytes is reduced to approximately 60% of cholangiocytes, which might indicate that Mdm2 sequence is floxed-out (**Supplementary Figure 2b**). This indicates that recombination had occurred in the cholangiocytes as a consequence of Cre activation after tamoxifen administration.

Legend: Quantification of *Mdm2* in cholangiocytes (in the *Mdm2*-RNAscope images) shows the loss of *Mdm2* after the induction of the model. ** denotes $p < 0.01$, (Mean \pm SEM) Student's t-test, (N=3 per group)

In hepatocytes, the expression of *Mdm2* remains similar with and without induction at different time points (see below, **Supplementary Figure 2d**). Therefore, the presence of senescent hepatocytes is interpreted as the consequence of paracrine senescence, and not as primary senescence induced by Cre recombination or spontaneous loss of *Mdm2*.

Legend: Quantification of *Mdm2* in hepatocytes (in the *Mdm2*-RNAscope images) shows that hepatocytes maintain similar levels of *Mdm2* after the induction of the model. Student's t-test. (N=3 per group). The blue line depicts the level of *Mdm2* for both groups.

Furthermore, the levels of *Mdm2* in cholangiocytes after induction (see **Supplementary Figure 2b**) closely relate to the levels of recombination found in the K19Mdm2 model (assessed by the percentage of TdTom-

positive cholangiocytes, see **Supplementary Figure 7c**, depicted here below), suggesting that the cholangiocytes that have lost *Mdm2* are a product of recombination after Cre-activation.

Legend: Primary senescence in cholangiocytes is similar in both models. Images show YFP-positive cholangiocytes (green) and tdTom-positive (red) K19-positive cholangiocytes (grey). Upper corner of the image shows a digital magnification of representative areas. Far right, quantification of the number of YFP per K19-positive cholangiocytes (OPN-Mdm2^{fllox/fllox}-YFP model) vs tdTom per K19-positive cholangiocytes (K19-Mdm2^{fllox/fllox}-tdTom^{LSL}). Images show uninduced (Cre+ -TM) versus induced (Cre+ +TM) mice at day 2 after last tamoxifen injection.

We think the reviewer may also be suggesting that our results could be explained by cholangiocytes deleting MDM2 without activating the TdTom tracer and then becoming hepatocytes that are senescent- we cannot plausibly see this as an explanation given that cholangiocytes do not become hepatocytes unless there is strong selection pressure from hepatocyte injury and impairment of hepatocyte replication in the context of highly proliferative cholangiocytes (Raven et al. Nature).

10. Fig. 3c and 3d. Was p27 detected in bile ducts? According to the conclusion by the authors, p53+ primary senescent cholangiocytes do not express p27, but only p53- secondary senescent cholangiocytes express p27.

Yes, p27 is expressed in bile ducts as well.

We have included PCR gene expression analysis of *cdkn1b* in isolated bile ducts at day 2 after tamoxifen induction (Observe *cdkn1b* expression in **Supplementary Figure 9b**).

Legend: qRT-PCR of cell-cycle related genes *cdkn1a* (p21), *cdkn1b* (p27) and *cdkn2a* (p16) show a significant increase in isolated bile ducts of Cre+ +TM (N=6) group, when compared with the Cre- +TM group (N=5) (day 2 after induction). ** denotes $p < 0.01$, *** denotes $p < 0.001$ (Mean \pm SEM). Mann-Whitney test.

As the SASP can also act in an autocrine manner to reinforce senescence in the cell-of-origin (in this case, the primary senescent cholangiocytes)¹⁻³, expression of p27 in cholangiocytes could be associated with TGF β autocrine signaling.

We have changed the text in the manuscript to make it clear. It now reads ‘

‘p27, a direct target of Transforming Growth Factor β (TGF β), is also expressed in cholangiocytes and hepatocytes in a similar pattern to p21 (Fig. 3c and 3d).’...

‘TGF β can induce senescence in a paracrine manner, via a mechanism that generates reactive oxygen species and DNA damage^{11, 17}. TGF β blocks the progression of the cell cycle at the restriction point in late G1 phase by transcriptional activation of cyclin-dependent kinase inhibitors (p21, p27 and p15,) through the SMAD complex^{11, 17, 18}.

1. Hoare M and Narita M. Transmitting senescence to the cell neighbourhood. Nature Cell Biology. 2013; 15, 887-889.
2. Coppé, JP, Desprez PY, Krtolica A, Campisi J. The senescence-associated secretory phenotype: the dark side of tumor suppression. 2010; 5:99-118.
3. Salama R, Sadaie M, Hoare M, Narita M. Cellular senescence and its effector programs. Genes Dev. 2014. 15; 28(2):99-114.

11. Fig. 3h

Activation of TGF-beta signaling in hepatocytes should be confirmed.

We have now included RT-PCR data for TGFb receptor I and II in hepatocytes in uninduced vs induced K19MDM2 model. Significant increase of both TGFbRI and TGFbRII indicate an active TGFb signaling response in hepatocytes (**Supplementary Figure 9d**).

Legend: qRT-PCR of *Tgfb1* and *Tgfb2* in isolated hepatocytes of Cre- +TM (N=5) and Cre+ +TM (N=6) mice at day 2 after induction are significantly increased. * denotes $p < 0.05$, ** denotes $p < 0.01$ (Mean \pm SEM). Mann-Whitney test.

We have also included in **Supplementary Figure 9e** in situ hybridization (ISH) for *Tgfb1*. In this images, *Tgfb1* is expressed in cholangiocytes as well as in some hepatocytes after the induction of the model (Cre+ +TM Day 2). ISH for *Tgfb1* reveals *Tgfb1* expression in macrophages as well. It would be interesting in the future to assess the role of macrophages in the transmission of senescence in biliary injury.

Legend: ISH by RNAscope for *Tgfb1* in Cre- +TM and Cre+ +TM at day 2 (N=3 per group). For the Cre- +TM, the selected areas show the absence of *Tgfb1* in cholangiocytes. For the Cre+ +TM, selected areas are magnifications of cholangiocytes (bottom left) and hepatocytes (bottom right) that express *Tgfb1*. Notice the presence of some *Tgfb1*-positive macrophages in the Cre+ +TM group. Right *PPIB* positive control (N=1). Scale bars = 50 μ m

12. Fig. 4. The mouse line shows spontaneous ductular reaction without DDC (Sup Fig. 2a) . Was DDC-induced ductular reaction enhanced or reduce?

We have now included in **Supplementary Figure 10f** a comparison between Day 90 after induction of the model (S Day 90), Oil+DDC and Tamoxifen+DDC in the model.

Assessment of K19 staining shows that DDC-induced ductular reaction is enhanced in comparison with the rest of the groups. This could reflect an inability to proliferate of the senescent cells in response to DDC diet at short term.

Increase of Ductular reaction at day 90 in the model could reflect the effects of paracrine senescence in the surrounding cholangiocytes at long term.

Legend: Comparison of the total percentage of K19-positive cells. From left to right: day 90 after induction of the K19-Mdm2^{flox/flox}tdTom^{LSL} model; C57BL6 DDC (1 week DDC in wild type C57BL6 mice), and S+DDC (induced K19-Mdm2^{flox/flox}tdTom^{LSL} maintained 1 week under DDC diet).

13. The statement (line 113) “We also found increased numbers of αSMA-positive cells surrounding the senescent 114 cholangiocytes (Fig. 2g and Supplementary Fig. 3)” is not appropriate, because the authors measured fluorescence intensity of αSMA surrounding the bile duct. For this claim, the cell number should be counted.

Thank you for pointing this out. We have now changed from “numbers” to “intensity”, and the legend in the figure 2g reads: “We found increased intensity of αSMA-positive cells surrounding the senescent cholangiocytes (Fig. 2g, h).

We have also included the percentage of αSma-positive cells in Figure 2h to give an idea of the increase of the response of αSma-positive cells in the whole tissue.

Legend: Increased total percentage of αSMA-positive cells in the Cre+ +TM group at day 21.

14. Fig. 6c and 6d. Specificity of YFP+ biliary cells on the effect of inhibition of TGF-beta signaling needs to be shown.

Quantification of senescence markers is performed exclusively on the YFP-positive biliary cells (with or without LY2157299 treatment). To clarify this point, we have changed the text to:

‘Quantifications in the YFP-biliary cells showed increased senescent markers and decrease BrdU incorporation, demonstrates that senescence can be transmitted in a paracrine manner *in vitro* (Fig. 6b).

In order to inhibit the transmission of paracrine senescence, we blocked TGFβ signaling pathway and assessed the same markers of senescence in the YFP-biliary cells. To that purpose, we used a small molecule that targets the TGFβ receptors ALK4, ALK5 and ALK7²³ (LY2157299), effectively blocking TGFβ signaling pathway (Fig. 6c).

Addition of LY2157299 caused a significant reduction of paracrine senescence in the YFP-biliary cells (Fig. 6d and Supplementary Fig. 7a, b),’

15. Fig. 7f. Effect of TGF-beta inhibitor. The authors state that the number of primary senescent cholangiocytes (tdTom-positive) remain unaltered, proving that primary senescence is not affected by the administration of LY2157299 (page 25, line 376) . However, tdTom+ cells were created by Cre-mediated recombination but was independent on senescence induction.

We wanted to indicate that administration of LY2157299 only blocks paracrine senescence. It is not designed

as a strategy to eliminate primary senescent cells (TdTom-positive). Whether primary senescence is affected by this compound, could be interesting to the field to investigate in the future. To clarify this point in the text, we have now changed to: `Addition of LY2157299 caused a significant reduction of paracrine senescence in the YFP-biliary cells (Fig. 6d and Supplementary Fig. 7a, b), without altering the number of primary senescent biliary cells (Supplementary Fig. 7c).

16. Page 11, line 182 [other members of the TGF-beta family, such as Tgfr1 and Tgfr2]
Tgfr1 and Tgfr2 are receptors, but not TGF-beta family members.

Thank you! We have now changed to: “After senescence, isolated bile ducts also exhibit increased gene expression of *Tgfr1* ($p=0.0023$) and *Tgfr2* ($p=0.0047$).”

17. In the *in vitro* system, macrophages removed senescent cells, whereas *in vivo* results (Fig. S2a) showed that tdTomato+ cells were not reduced. This discrepancy needs explanation.

Currently there is a trend to decrease in the number of total TdTom-positive recombined senescent cells in the *in vivo* model (see **Supplementary Figure 3a**). This result suggest that the recruited macrophages may actively target and eliminate senescent cells.

Legend: Quantification of the total percentage of TdTom-positive cells increases after induction of the model at day 2 but shows no significant differences over the course of 90 days. ANOVA, Sidak’s multiple comparisons test.

We do observe infiltration of macrophages at day 5 and accumulation of mononuclear infiltrates at day 21. The elimination of senescent cells by immune surveillance has been described as a fast process in tumoural context^{1,2}. However, in our model, this period might be longer (as the mechanisms quite significantly differ). Alternatively, macrophages might target and engulf the apoptotic cells present in the model and then target the senescent cholangiocytes.

Our *in vitro* system represents a conceptualization of this process, in which we established a direct co-culture of senescent cholangiocytes and macrophages. This could potentially explain the fast target and elimination of senescent cholangiocytes (72 hours).

1. Kang et al. Senescence surveillance of pre-malignant hepatocytes limits liver cancer development. *Nature*. (2011). 479(7374): 547-51
2. Xue et al. Senescence and tumour clearance is triggered by p53 restoration in murine liver carcinomas. *Nature* (2007). 445(7128): 656-60.

REVIEWER 4.

The authors generated a model of biliary senescence by conditionally knocking out Mdm2 in bile ducts. Senescent cholangiocytes induced profound alterations in the microenvironment with recruitment of myofibroblasts and macrophages causing collagen deposition, TGFbeta production and induction of senescence in surrounding cells. This new model supports the concept of paracrine transmission of senescence. The authors identified TGFbeta as a major mediator of this response.

The in vitro and in vivo experiments are very convincing and provide a rationale for targeting TGFbeta to treat biliary disease and accompanying liver damage.

Comments:

1. The TGFbeta type I receptor (ALK5) inhibitor used in the study (LY2157299) is not specific as it will also block signaling through ALK4/ALK7 - this is an important point to include.

Thank you for pointing this out. We have changed the phrase to: "we used a small molecule that targets the TGFβ receptors ALK4, ALK5 and ALK7²³ (LY2157299) to block TGFβ signaling pathway".

2. In fact, it would be great for the field if the TGFbeta ligand(s) contributing to the paracrine transmission of senescence could be identified. The authors show that TGFbeta1 mRNA levels are elevated in their model, but what about TGFb2 and TGFb3, which utilise the same signalling pathway? Or activin A and activin B, which utilise ALK4/ALK7 and have previously been implicated in cholangiocyte development and liver regeneration. Showing which of these ligands contributes to senescence would be a major step forward, as it would enable the utilisation of more specific inhibitors.

We do completely agree. When we initially tested the values of TGFbeta family members (such as *Tgfb2* and *Tgfb3*) there was a non significant trend to increase ($p=0.3203$ for *Tgfb2* and $p=0.4156$ for *Tgfb3*).

We have now included these previous data to **Supplementary Figure 9c**. We have also performed a new PCR for Inhibin A and B, which were non-significant as well.

Legend: qRT-PCR of other members of the TGFb signaling pathway, such as *Tgfb2*, *Tgfb3*, *Inha* (Inhibin a) and *Inhb* (Inhibin b) show no significant differences in isolated bile ducts of Cre+ +TM (N=6) group, when compared with the Cre- +TM group (N=5) (day 2 after induction). Mann-Whitney test.

We have now also included in **Supplementary Figure 9b** data about the expression of cell cycle genes such as *cdkn1a* (p21), *cdkn1b* (p27) and *cdkn2a* (p16) as positive controls to show that cell cycle stop is, in fact, established in isolated bile ducts in induced mice (Cre+ +TM) vs non-induced (Cre- +TM) at day 2 post-tamoxifen administration. We hope this will help to reinforce the idea that our model presents senescent cholangiocytes after tamoxifen induction.

Legend: qRT-PCR of cell-cycle related genes *cdkn1a* (p21), *cdkn1b* (p27) and *cdkn2a* (p16) show a significant increase in isolated bile ducts of Cre+ +TM (N=6) group, when compared with the Cre- +TM group (N=5) (day 2 after induction). ** denotes $p < 0.01$, *** denotes $p < 0.001$ (Mean \pm SEM). Mann-Whitney test.

We have also included in the **Supplementary Figure 9a** PCR controls for the isolation of bile ducts and hepatocytes in Cre- +TM mice at day 2 after tamoxifen induction. Isolated bile ducts expressed Keratin 19 (K19) but low levels of HNF4a. Conversely, isolated hepatocytes express HNF4a but no K19. We hope these data will give an idea about the purity of the populations isolated (as explained in the Methods section).

Legend: qRT-PCR analysis of *HNF4a* and *K19* in the isolated bile ducts (N=5) and hepatocytes (N=5) of Cre- +TM and Cre+ +TM mice day 2 after induction. Hepatocytes have high expression of the hepatocyte marker *HNF4a* but low expression of the epithelial marker *K19*. On the other hand, Bile ducts have low expression of *HNF4a* and high expression of *K19*. * denotes $p < 0.05$, ** denotes $p < 0.01$ (Mean \pm SEM). Mann-Whitney test.

3. If other TGFbeta ligands are upregulated, then their importance could be examined in the in vitro model. Please, refer to previous answer.

4. In Fig. 7B, it is difficult to see the p21+ cells - can these be expanded?

We have now taken new representative images magnification. We have also included an isotype control which we completely forgot to add previously. Hopefully this will help with the identification of the p21+ cholangiocytes.

Legend: Representative images of p21 (red) in cholangiocytes (green) in mice treated with vehicle, 10 or 20 mg/kg of LY2157299. (c) Percentage of p21-positive cholangiocytes diminishes with the administration of LY2157299.

5. Fig 6A & C - Should read "Quantify" and "Quantification" (in the actual figure)

We have now changed from Cuantify and Cuantification to Quantify and Quantification- many thanks!

Reviewers' Comments:

Reviewer #1:

Remarks to the Author:

The authors have addressed all of my concerns and have incorporated all of the data into the manuscript as suggested.

Reviewer #2:

Remarks to the Author:

The authors have carried out a most thorough response to the questions raised by each of the 4 referees

Referee 1: they have looked at senescence in other K19 tissues, though the micrograph states p21, the legend states p16?? They have also responded to the query regarding disease progression and looked at Ki67 and p21 in controls.

Referee 2: Explained apparent inconsistencies between quantitative data and Figs due to random selection of Figs and high inter-animal variability.

Referee 3: Justification for focussing on TGF β is given, along with acknowledgement that other factors may be involved in the SASP. Showed that proliferation arrest was irreversible, at least for up to 90 days. TUNEL staining showed that apoptosis was probably a secondary consequence, but do acknowledge that apoptosis may amplify the effect of senescence. Also exclude possibility that WT-1 is involved. Isotype controls now included for DCR and H2a immunofluorescence. Also show relative expression of mdm2 in cholangiocytes and hepatocytes, showing mdm2 is floxed out of many cholangiocytes, while mdm2 expression in hepatocytes remains very high with/without induction by Tm, suggesting recombination does not happen in hepatocytes. There was no evidence that cholangiocyte to hepatocyte metaplasia occurred, moreover it was considered highly unlikely that cholangiocytes could delete mdm2 without activating the TdTom marker and then undergo metaplasia. The TGF β signalling pathway was also confirmed by expressions of TGF β 1 and TGFR1 and R2.

Referee 4: The limitations (lack of specificity) of the inhibitors was acknowledged along with data on expressions of other TGF family members, though none appeared to be involved in this model. Better p21 images have been provided

Reviewer #3:

Remarks to the Author:

The authors have addressed my comments adequately. I have no further comments.

Reviewer #4:

Remarks to the Author:

The authors have addressed all of my concerns. In addition, they appear to have gone to considerable effort to answer the questions posed by the other reviewers. I think the manuscript is better for the new additions/clarifications. I recommend the article be published in Nature Communications.

Dear Reviewers- we are grateful for the careful review of our work. We are pleased with the enthusiasm shown for this manuscript and the constructive feedback. We have addressed the points below.

REVIEWER 1.

Sofia Ferreira-Gonzalez et al., present a well-written manuscript which introduces a novel mouse model of biliary epithelial cell senescence. This model system will be a useful tool in determining the functional role of biliary epithelial cell senescence in the pathogenesis of the cholangiopathies, including Primary Sclerosing Cholangitis (PSC) and Primary Biliary Cholangitis (PBC).

Their model system utilizes an inducible Cre-Lox system to knockout MDM2, a negative regulator of p53, selectively in cholangiocytes. This system successfully induced senescence in the biliary epithelia and the surrounding parenchyma through SASP (TGFbeta). They showed that senescent cholangiocytes could exaggerate liver disease observed in a mouse model of PSC (DDC-diet). They also showed that senescent cholangiocytes negatively impacted the regenerative capacity of the liver after partial hepatectomy. While the authors present a strong manuscript, the following should be addressed prior to publication.

The weaknesses of this manuscript are:

(i) While their inducible Cre-Lox system successfully induced cellular senescence in the liver, the authors did not examine other K19 positive epithelia. K19 is expressed in various epithelia, including those in the intestine and kidney. Therefore, at a minimum, the authors need to examine these tissues for senescence and fibrosis. Thank you for pointing this out. We have analyzed other K19-positive tissue such as Kidney and Gut (see **Supplementary Figure 6**). As expected, both of them present p21-senescent epithelial cells. Fibrotic deposition is exacerbated as time passes (compare Picosirius red accumulation at day 2 and day 90).

From left to right: Left: Cre-negative mice at day 2 after tamoxifen administration. Center: Cre-positive mice at day 2 after tamoxifen administration. Right: Cre-positive mice at day 90 after tamoxifen administration.

(a) p16 immunohistochemistry in Cre- +TM at day 2 shows low expression of p16 in kidney and gut epithelia. After induction, p16 is expressed in both epithelia at day 2 and day 90. **(b)** Collagen accumulation assessed by PiSR staining shows increased deposition over the course of time in gut and kidney epithelia. Scale bars = 50µm.

(ii) The authors neglected to look at the p16 signaling pathway, a major driver of cellular senescence. It is possible that p16 is not involved in driving biliary senescence in their model. However, the authors need to examine and discuss it. Are there p16 positive cholangiocytes? Are there any p16 positive cells in the parenchyma? Do isolated bile ducts overexpress p16?

Thank you for this observation. We have included immunohistochemistry for p16 in our model (**Supplementary Figure 4a**). After induction of senescence, p16 is accumulated in cholangiocytes. Furthermore, p16-positive hepatocytes are found in the parenchyma in close proximity to the p16-positive cholangiocytes at day 21 and day 90 post-induction.

Legend: p16-immunohistochemistry shows accumulation of p16 positive cells in cholangiocytes in Cre+ +TM mice at day 2 after induction. Some hepatocytes express p16 as well (red arrows, areas surrounded in red). From left to right: Left: Cre-negative mice at day 2 after tamoxifen administration. Center: Cre-positive mice at day 2 after tamoxifen administration. Right: Cre-positive mice at day 90 after tamoxifen administration. N=4-6 mice per group.

Supporting this result, we found that the expression of *cdkn2a* is significantly increased in isolated bile ducts ($p=0.0006$), **Supplementary Figure 9b**).

Legend: Isolated bile ducts express *cdkn2a* in induced mice (Cre+ +TM) when compared with un-induced mice (Cre- +TM) at day 2 after tamoxifen administration.

Furthermore, use of TGFb inhibitor (LY2157299) in the setting of Senescence+DDC in our K19MDM2 model results in a significant decrease of total p16 expression (**Figure 7.I**)

Legend: Total percentage of p16-positive cells diminishes with the administration of LY2157299 in the K19-MDM2 model after tamoxifen-induction and 1 week of DDC diet. N=5 mice per group.

(iii) The authors extended their mice cohort for 90 days. They showed that senescent cholangiocytes remained viable. However, they did not examine liver disease progression. It would be predicted that these senescent cholangiocytes continue to produce SASP, which negatively impacts liver function. The authors need to check the biochemistry of the liver (ALP, ALT) and fibrosis.

We have analyzed total percentage of PicroSirius red staining in induced mice (Cre+ +TM) at day 2, day 21 and day 90 after induction. Results show a significant increase of fibrotic response as time passes (see **Supplementary Figure 5d**).

Legend: Quantification of collagen deposition assessed by PiSR staining over the course of 90 days after the induction of the model. * denotes $p < 0.05$, ** denotes $p < 0.01$, (Mean \pm SEM). ANOVA, Sidak's multiple comparisons test. N=4-6 per group.

Furthermore, the significant increase of transaminases at day 90 in comparison with day 2 after induction, suggest that liver function is further impaired as time passes (see **Supplementary Figure 3d**).

Legend: Liver transaminases in healthy control, Cre- +TM at day 2 and Cre+ +TM at day 2, 21, 60 and 90 after tamoxifen administration. Significant increase of ALT, AST and Bilirubin at day 90 compared with day 2 post-induction. The rest of transaminases (ALB, ALP and Cholesterol showed a trend to increase). N=4-6 per group.

We have also now included ductopenic index over the course of time in **Supplementary Fig. 3e**. There are no significant changes in the number of bile ducts per portal tract over the course of 90 days.

Legend: Ductopenic index shows no significant alterations in the course of 90 days. The blue line depicts the baseline level of ductopenia. ANOVA, Sidak's multiple comparisons test. (N=4-6 per group).

Minor comments:

In figure 2: There is a lack of consistency in labels. Some IF panels have the precise date after induction, while others don't have any label. This made it challenging to interpret results.

Thank you for pointing this out. We have now included the date after induction in all the panels of this figure, which will hopefully help to understand these results.

In figure 3: The pSmad2/3 is not convincing. It looks like only one mouse has an increase (#4). Additionally, an increase in pSmad2/3 is evident in some of the Cre- +TM. The authors need to repeat or elaborate on this finding.

Thank you for pointing this out. We have changed the quantification data in **Figure 3h** from a column to a scatter dot-plot, to help to identify the levels of NFkB and pSmad2/3 in the isolated bile ducts per mouse (normalized to bActin).

From this data it become obvious that:

1. Some Cre- +TM isolated bile ducts express basal levels of pSmad2/3 and NFkB
2. There is a high variability in the Cre+ +TM group, which could indicate that not all bile ducts respond to tamoxifen in the same manner. This variability also correlates with the variability of Cre-dependent recombination (assessed by TdTom response per bile duct) as seen in **Supplementary Figure 7c**. Recombination rates in our model range from 30 to 68%, which could explain the variability in the senescent response.

We acknowledge that there some animals present pSmad2/3 and NFkB activation in the Cre- +TM group. This result is consistent with the gene analysis data from **Figure 3f, g**, where the baseline levels in the Cre- +TM control show activation of *Tgfb* and pro-inflammatory cytokines such as *Il6*.

Furthermore, the newly added RNA Scope for *Tgfb1* in Cre- +TM mice at day 2 after tamoxifen induction shows some *Tgfb1*-positive cholangiocytes and hepatocytes (**Supplementary Figure 9e**). This data supports this activation of pSmad2/3 signaling pathway in Cre- mice (see image below).

Legend: ISH by RNAscope for *Tgfb1* in Cre- +TM and Cre+ +TM at day 2 (N=3 per group). For the Cre- +TM, the selected areas show the absence of *Tgfb1* in cholangiocytes. For the Cre+ +TM, selected areas are magnifications of cholangiocytes (bottom left) and hepatocytes (bottom right) that express *Tgfb1*. Notice the presence of some *Tgfb1*-positive macrophages in the Cre+ +TM group. Right *PPIB* positive control (N=1). Scale bars = 50µm

We have noticed that tamoxifen administration causes some degree of biliary damage (see the levels of bilirubin in the image below), which could potentially explain the activation of these signaling pathways in our model. In fact, some studies already pointed out a direct relationship between the use of tamoxifen and hepatotoxicity¹⁻⁴

Legend: Bilirubin levels in C57BL6J Healthy mice and mice administered with 1, 2 or 3 intraperitoneal doses of tamoxifen in alternate days increases biliary injury (we consider pathological levels of bilirubin above 10 umol/l, See Supplementary Figure 3d). * denotes $p < 0.05$, ** denotes $p < 0.01$, (Mean \pm SEM). ANOVA, Sidak's multiple comparisons test.

1. Floren LC et al. Tamoxifen in liver disease: potential exacerbation of hepatic dysfunction. *Ann Oncol.* 1998 9(10): 1123-6.
2. Gao FF et al. Tamoxifen induces hepatotoxicity and changes to hepatocyte morphology at the early stage of endocrinotherapy in mice. *Biomed Rep.* 2016. 4(1):102-106.
3. De Conti A et al. Genotoxic, epigenetic and transcriptomic effects of tamoxifen in mouse liver. *Toxicology.* 2014. 325: 12-20.
4. Cole, LK; Jacobs, EL and Vance, DE. Tamoxifen induces triacylglycerol accumulation in the mouse liver by activation of fatty acid synthesis. *Hepatology.* 2010 52(4): 1258-65.

In figure 4: The authors should add a WT group and/or a vehicle treated group to the figure panels. DDC treatment alone will increase serum biochemistries compared to untreated or vehicle treated. It would be beneficial to know how DDC, compared to untreated or vehicle control, alters Ki67 and p21 expression.

We have now included in **Supplementary Figure 10a, b, c, d** and **e** a comparison between:

- **C57BL6 Healthy**: Healthy WT-C57BL6/J group (the same background as the our K19-MDM2-TdTom model).
- **C57BL6 DDC**: 1 week DDC in WT-C57BL6/J group.
- **K19Mdm2 Oil + DDC**: 1 week DDC in the uninduced K19-MDM2-TdTom model (administered with the vehicle, oil). This is the same group of Figure 4).

As expected there is an increase of Ki67 and p21 with DDC diet when compared with the healthy group. This is also the case for transaminases' levels. We haven't found significant differences between DDC and Vehicle+DDC groups (see below).

Legend: (a) Liver transaminases (alanine transaminase, aspartate aminotransferase, cholesterol and bilirubin). (b) Total percentage of proliferating (Ki67-positive) K19-positive cells. (c) Total percentage of senescent (p21-positive) K19-positive cells. (d) Total percentage of proliferating (Ki67-positive) cells. (e) Total percentage of senescent (p21-positive) HNF4a-positive hepatocytes.

Other remarks:

The authors should carefully review the references. The citations in the body of the text have a few mistakes. For example, page number 28, reference number 29 is incorrect.

Thank you very much for this careful observation (we skipped one number in the text). We have now fixed this and, for example, reference 29 is now Tabibian et al., 2016 (which is actually a seminal paper in senescence and biliary disease).

REVIEWER 2.

This is a well written and illustrated paper that shows that biliary cell senescence can be a driving force for biliary damage and that senescence can also exacerbate other biliary injuries. The paper combines both in vitro and in vivo descriptive and functional data to show a pivotal role for TGFbeta in the transmission of a detrimental senescent phenotype to neighbouring parenchymal (hepatocyte) cells, and how this pathway may be targeted for disease alleviation. I have a few points that are dealt with in order

1. Figure 1 mentions 'Explants'. As far as I can see these are resections or biopsy specimens. An Explant usually refers to a tissue removed from the body and cultured in a medium.

Thanks for pointing this out. In the methods sections we wrote biopsies. We have now changed to biopsies in the text as well.

2. Page 7 mentions 'a large polymorphonuclear infiltrate identified as F4/80-positive macrophages' Polymorphonuclear' refers to the likes of neutrophils and eosinophils., replace with 'mononuclear infiltrate'

We do completely agree. We have changed to mononuclear infiltrate.

3. Please explain some seemingly inconsistencies between Figures. In Fig. 2c and 3a there are generally few p53+ cholangiocyte nuclei, whereas much more abundant in 3e.

For the manuscript, we tried to include representative images of each immunofluorescence, which sometimes may not match the quantitative data provided.

In the K19-Mdm2 model we have found a high variability in the Cre-dependent recombination response (ranging from 31 to 68% of recombination, as seen in **Supplementary Figure 7.c, right**). This could potentially account for the apparent differences in the number of p53+ cholangiocytes in the different images.

Legend: Primary senescence in cholangiocytes is similar in both models. Images show YFP-positive cholangiocytes (green) and tdTom-positive (red) K19-positive cholangiocytes (grey). Upper corner of the image shows a digital magnification of representative areas. Far right, quantification of the number of YFP per K19-positive cholangiocytes (OPN-Mdm2^{flox/flox}-YFP model) vs tdTom per K19-positive cholangiocytes (K19-Mdm2^{flox/flox}-tdTom^{LSL}). Images show uninduced (Cre+ -TM) versus induced (Cre+ +TM) mice at day 2 after last tamoxifen injection.

Fig. 3e, why are the p53+ bile ducts (green) not expressing the likes of 53BP1 and DCR2 as shown in 2d?

We apologize for the quality of **Figure 3e**, the exposure of p53 in the green channel covers the expression of 53BP1, H2A.X and DCR2 (red). In the original images, p53+ bile ducts express 53BP1, DCR2 and H2A.X. We have included in **Supplementary Figure 8a** single stainings of **Figure 3e** to prove the expression of these markers in the bile ducts. For p53+DCR2 double immunofluorescence, as DCR2 expression is cytoplasmatic, we have included an area of extrahepatic bile ducts (where cholangiocytes have a higher cytoplasm/nuclei ratio) so the staining is more clear.

a

Legend: Single channels for Figure 3e (Fig 3. Paracrine senescence in the model is TGF β -dependent). Double immunostaining for p53 (green) and several senescent markers in Cre+ +TM mice at day 2 after induction (N=6 per group). From left to right: 53BP1, H2A.X and DCR2 (red). From up-down: Merge, p53, senescent marker, DAPI. Scale bars = 50 μ m.

There seems to be a huge discrepancy in the numbers between those hepatocytes expressing p27 (very few in 3c) and those expressing 53BP1, H2A.X and DCR2 (most hepatocytes in 3e).

To explain this discrepancy, we have quantified H2A.X, DCR2, 53BP1, p21 and p27 in hepatocytes (included in **Supplementary Figure 8b**).

While there is a high variability between animals, these data suggest that the paracrine response in the parenchyma is similar in terms of expression of p27, p21 or the different senescent markers (53BP1, DCR2 and H2A.X). The images used in the manuscript are intended to be representative images. However, the animals depicted are selected randomly for each image, increasing the variability in between figures.

4. The labeling of many of the quantitative data graphs is ambiguous. For example, 4e, j, and l, 5c and 5e, 7d, e, k and l all have units expressed as e.g. % F4/80/field or %Ki67/field. As far as I can tell these are %s calculated from counting cells in 30 fields at 200x magnification (Methods. 6. Microscopy and cell counting). Therefore, the word 'field' is redundant and will confuse readers in to thinking this has something to do with counting per HPF.

We do agree. Quantifications are made by counting cells in 30 fields per slide, and adding "field" to the graph axis only adds more confusion. We have just indicated the percentage and eliminate the word field from the Y axis of the figures.

5. Page 18 bottom line "through a TGFb-dependent SASP mechanism' The authors are being premature in this statement. The functional data proving this are introduced after this statement.

We do agree, the role of TGFb in the system is disclosed in the *in vitro* experiment in the next figure. We have changed to: "These results suggest that cholangiocyte senescence impairs the regenerative response of the liver parenchyma. This could potentially account for the loss of hepatocyte function seen in human PSC/PBC patients."

In the next section we introduce the potential relevance of TGFb in the model with:

"To test whether transmission of senescence from cholangiocytes to the surrounding hepatocytes is TGFb-dependent, we generated an *in vitro* model based..."

REVIEWER 3.

Relationship between cellular senescence and cholangiopathy has been known and there are several recent papers describing the mechanism. It has been also shown that SASP including TGFb derived from bile ducts in senescence affect surrounding cells. Thus, conceptual advance by this paper may be limited. However, previous reports were mostly based on in vitro experiments. In this paper, the authors used their own in vivo model to demonstrate causal relationship. The model can be used to manipulate senescence in vivo and would be useful.

1. It is not clear why the authors focused on TGF-b among various SASP and whether TGF-b derived from bile ducts directly affects bile ducts and hepatocytes.

We initially focused into TGFb signaling as, in our model, *TGFb1* presented the highest significant variance from all the assessed SASP's factors, i.e. *TGFb1* ($p=0.0007$) vs *Nfkb* ($p=0.0132$), *Il1a* ($p=0.0160$) or *Il6* ($p=0.1708$).

Expression of p27, a direct target of TGFb signaling pathway¹ in hepatocytes (see **Figure 3c, d** and **Supplementary Figure 9b**) suggest a direct functional role for TGFb in the expansion of paracrine senescence response.

A functional proof of the role of TGFb in our model arises from the use of LY2157299 which directly blocks TGFb signaling pathway (*in vitro* in **Figure 6** and *in vivo* in **Figure 7**). The use of this small molecule in our K19-MDM2 model (**Figure 7**) decreased the levels of paracrine senescence in the parenchyma and proved the role of TGFb in the transmission of senescence to the liver parenchyma.

We do acknowledge that TGFb is just one single factor in a myriad of SASP components. At the time of these experiments (and taking into consideration the importance of TGFb in the transmission of the SASP, as shown by Acosta and colleagues²) we considered the use of TGFb inhibitors to partially stop the spreading of paracrine senescence. While this approach partially blocks the effects of senescence, it improves features of biliary injury in our model, such as fibrosis. Furthermore, it proves the importance of TGFb in senescence-related biliary injury.

1. Muñoz-Espín D, Serrano M. Cellular senescence: from physiology to pathology. *Nat Rev Mol Cell Biol.* 2014; 15(7): 482-496.
2. Acosta, JC. et al. A complex secretory program orchestrated by the inflammasome controls paracrine senescence. *Nat Cell Biology.* 2013; 15(8): 978-990.

2. The authors used floxed K19-Mdm2 and liver specific tdTom mice, and show expression of cellular senescence markers. However, it is not clear whether cell cycle arrest was irreversible or transient. Additional evidence would be necessary.

We have now included a quantification of proliferating (Ki67-positive) recombined (TdTom-positive) cells in **Supplementary Figure 2e**. This figure shows that percentage of recombined (senescent cholangiocytes) is very low and does not vary with the course of time (final time point 90 days), suggesting that cell cycle arrest is irreversible.

Legend: Quantification of proliferation (assessed by Ki67) in TdTom-positive cholangiocytes shows no significant differences over the course of 90 days. ANOVA, Sidak's multiple comparisons test. N=4-6 per group.

3. In this mouse model, not only senescence but also apoptosis was induced (Sup Fig. 2d) . Thus, it is not clear whether aggravation of liver injury was due to senescence, apoptosis, or both.

We have calculated the number of TUNEL-positive cholangiocytes in our model up to day 90 post-induction (see **Supplementary Figure 4d**). Number of TUNEL-positive cells appear to increase at day 21 and 90 compared

with day 2, suggesting a progressive course for apoptosis. This late appearance of TUNEL-positive cells might indicate that apoptosis is a secondary consequence and/or an off-target effect of the induction of senescence

in our model.

Legend: TUNEL staining (red) shows an increase in apoptotic cholangiocyte nuclei in the Cre+ mice 21 days after induction. N= 4-6 per group. **(d)** Quantification of TUNEL-positive cholangiocytes over the course of 90 days after induction of the model. The image includes isotype control and DNaseI-positive control. ** denotes $p < 0.01$, (Mean \pm SEM). ANOVA, Sidak's multiple comparisons test. N= 4-6 per group. Scale bars = 50 μ m.

We do agree that, in our model, apoptosis can potentially amplify the effect of senescence to exacerbate biliary injury as has been previously postulated¹. We have added this data and comment to the paper.

1. Sasaki and Nakanuma. Biliary epithelial apoptosis, autophagy and senescence in primary biliary cirrhosis. *Hepat Res Treat.* 2010; 205128.

4. Recombination by K19-CreERT is not specific to bile ducts because K19 is expressed in mesothelial cells. Senescence of mesothelial cells needs to be examined. As mesothelial cells express various growth factors for hepatocytes, regeneration after PHx could be affected by mesothelial cells.

To evaluate the role of senescence in mesothelial cells in our model, we assessed the expression of a well known mesothelial marker: Wilms Tumor-1 (WT1) (Li, Wang and Asahina, 2013)¹.

In our model, WT1 is not expressed after induction of senescence (**Supplementary Fig. 6c**). Similarly, assessment at long term revealed no WT1 expression.

Legend: Immunohistochemistry for Wilms Tumor 1 (WT1) shows no involvement of mesenchymal cells in our model. This image includes isotype control and a positive control (Glisson Capsule of K19-Mdm2^{flox/flox}tdTom^{LSL} mice, connective tissue septa). Scale bars = 50µm.

In 70% partial hepatectomy (PH, with and without senescence) in which active proliferation of parenchymal cells occurs, we did not observe WT1 expression (**Supplementary Fig. 11d**). This result backups previous data presented by *Perugorria and colleagues in 2009*, in which WT1 is not up-regulated in the context of PH².

Legend: Immunohistochemistry for Wilms Tumor 1 (WT1) shows no involvement of mesenchymal cells in our model after partial hepatectomy. This image includes isotype control and a positive control (Glisson Capsule of K19-Mdm2^{flox/flox}tdTom^{LSL} mice, connective tissue septa). Scale bars = 50µm

Altogether, these data suggest that mesothelial cells do not play an active role in our model and that regeneration after PH is, in this case, not affected by mesothelial cells.

1. Li, Y., Wang, K., Asahina, K. Mesothelial cells give rise to hepatic stellate cells and myofibroblasts via mesothelial-mesenchymal transition in liver injury (2013). PNAS 110(6): 2324-9.
2. Perugorria, MJ. et al. Wilms' Tumor 1 gene expression in hepatocellular carcinoma promotes cell dedifferentiation and resistance to chemotherapy (2009). Cancer Res 69(4):1358-67.

5. There appear to be many hepatocytes with DCR2 and H2a. It is not clear whether they were back ground or real signals. Isotype data should be presented.

Thank you for pointing this out. We have included isotype data for H2A.X as well in **Figure 1c, d**.

Legend: PSC and PBC show DCR2 expression (green) in cholangiocytes (red). Far right, quantification. **(d)** PSC and PBC show γ H2A.X expression (green) in cholangiocytes (red). Far right, quantification. ** denotes $p < 0.01$, *** denotes $p < 0.001$, **** denotes $p < 0.0001$ (Mean \pm SEM). ANOVA, Sidak's multiple comparisons test. Scale bars = 50 μ m.

We have also included images of the liver parenchyma in human PBC, PSC explants in **Supplementary Figure 1a, b**. This images indicate that H2A.X and DCR-positive hepatocytes are quite common in biliary disease.

Legend: **(a)** H2A.X (green) is present in K19-cholangiocytes (red) and surrounding hepatocytes in PSC and PBC. Right, same images without DAPI. **(b)** DCR2 (green) is present in K19-cholangiocytes (red) and surrounding hepatocytes in PSC and PBC. Right, same images without DAPI.

In a previous publication by Sasaki and colleagues¹, p21-positive hepatocytes can be observed in late stages of PBC (see the image on the right). Although they did not elaborate on this finding, the fact that in our batch of samples some hepatocytes present markers of senescence (such as p16, DCR2 and H2A.X) reinforce this finding.

This suggest that in late stages of PBC and PSC, the senescence footprint expands from cholangiocytes into the liver parenchyma, exacerbating biliary disease.

1.Sasaki et al. Frequent cellular senescence in small bile ducts in primary biliary cirrhosis: a possible role in bile duct loss. *J Pathol* (2005). 205(4):451-9.

This figure represents p21 staining in PBC human section and belongs to **Figure 2 Panel F** in *Sasaki et al. 2005*.

6. Fig. 2c. The authors state that Ki67+ cells in p53+ cells were 0.1%. However, the data seem to show more Ki67+ cells. Are they p53 negative? More clear data should be provided. By staining K19, Ki67, and p53, Ki67 positive fraction in p53+ and p53- cells should be quantified.

Yes, only 0.1% of p53-positive cells are Ki67-positive.

We have now included in **Supplementary Figure 2c** the percentage of proliferating cells per bile duct in the p53-positive and p53-negative fractions at Day 2 post-induction of the model.

This result show that p53-positive cholangiocytes have a very low proliferative rate (with an average of 0.1% Ki67+ p53+), whereas p53-negative cholangiocytes have a significant increased proliferate capacity.

Legend: Quantification of the total percentage of p53-positive and p53-negative cholangiocytes that proliferate in the Cre+ +TM mice at day 2. This result shows that p53-positive cells do not proliferate while p53-negative cells do. *** denotes $p < 0.001$, (Mean \pm SEM) Student's t-test, (N=6 per group).

As Mosteiro et al¹ and Ritschka et al² confirmed, senescent cells induce profound alterations in their environment that are key to induce tissue regeneration and cellular plasticity. This could be potentially the case in our model, in which, for example the cholangiocytes or the Hepatic Progenitor Cell compartment could be proliferating in response to this senescent cues. However, this is highly speculative and further experimentation would be required to test this hypothesis.

1. Mosteiro, L. et al. Tissue damage and senescence provide critical signals for cellular reprogramming in vivo. *Nat Science*. 2016; 354 (6315).
2. Ritschka, B. et al. The senescence-associated secretory phenotype induces cellular plasticity and tissue regeneration. *Genes Dev*. 2017. 31(2):172-183.

7. Fig 2d. The authors state in text that cellular senescence was induced in cholangiocytes. However, the cell should be hepatocyte.

We wanted to express that senescence is initially established in cholangiocytes. However, as Reviewer 3 pointed out, there are some hepatocytes in this figure that express markers of senescence. As shown in following figures, this might suggest the presence of paracrine-senescence mechanisms from cholangiocytes towards the liver parenchyma. To clarify this point, we have now changed the text to:

'Expression of other senescence markers like 53BP1, γ H2A.X and DCR2 (Fig. 2d), suggest the occurrence of cellular senescence in cholangiocytes. Furthermore, some hepatocytes also express 53BP1, γ H2A.X and DCR2 (Fig. 2d), suggesting the presence of paracrine activity from the senescent cholangiocytes towards the liver parenchyma.

8. Supplementary Fig. 2a

The authors state that bile duct senescence induced ductular reaction but tdTom+ cholangiocyte senescence was not altered (p7). As ductular reaction accompanies cholangiocyte proliferation, the relative tdTom+ fraction is expected to decrease. But it was not decreased, suggesting that tdTom+ cholangiocytes were not irreversibly arrested.

We have quantified again our data on K19-staining and TdTom staining and presented it in the following manner:

- Total Percentage of K19-positive cells (**Supplementary Figure 3, a**). This would hopefully give an idea of the increasing ductular reaction seen in the model (as shown by the significant increase at day 90 compared with day 2 post-tamoxifen administration).

- Total Percentage of TdTom-positive recombined cells (**Supplementary Figure 3, b**) This figure shows that at 90 days post tamoxifen-induction there is a non-significant trend to decrease in total number of senescent cells ($p=0.3823$), suggesting that our model establishes an irreversible cell cycle arrest.

* **See answer 2.**

Legend: (a) Quantification of the total percentage of TdTom-positive cells increases after induction of the model at day 2 but shows no significant differences over the course of 90 days. ANOVA, Sidak's multiple comparisons test. (b) Quantification of the total percentage of K19 significantly increases over the course of 90 days. ** denotes $p < 0.01$, (Mean \pm SEM). ANOVA, Sidak's multiple comparisons test. N=4-6 per group.

9. Supplementary Fig. 4e and 4f and Fig. 3a

Based on p21+/tdTom- or YFP- hepatocytes and p21+/p53- cholangiocytes, the authors argue senescence in a paracrine manner. However, there are many other possibilities to be considered. For example, without recombination of tdTom or YFP, cholangiocytes with recombination of the *Mdm2* locus. Such cells could be converted to hepatocytes. Presence of cholangiocytes that were induced to express p21 by loss of *Mdm2* and p53 expression. Expression of *Mdm2* should be examined in the senescent cells as a result of paracrine manner.

We have performed in situ hybridization (ISH) for *Mdm2*. In control mice (Cre- +TM), *Mdm2* shows a homogeneous pattern of expression in both cholangiocytes and hepatocytes (**Supplementary, Figure 2a**).

Legend: ISH by RNAscope for *Mdm2* in Cre- +TM (N=3) and Cre+ +TM (N=3) at day 2. For the Cre- +TM, the selected area shows the presence of *Mdm2* in cholangiocytes. For the Cre+ +TM, representative images of the three mice (N=1, 2 and 3) show the loss of *Mdm2* in cholangiocytes (in N=1) while hepatocytes still maintain *Mdm2* expression (as seen in the magnified area of N=2). Right PPIB positive control (N=1). Scale bars = 50µm

After induction (Cre+ +TM at day 2, 21 and 90), *Mdm2* expression in cholangiocytes is reduced to approximately 60% of cholangiocytes, which might indicate that *Mdm2* sequence is floxed-out (**Supplementary Figure 2b**). This indicates that recombination had occurred in the cholangiocytes as a consequence of Cre activation after tamoxifen administration.

Legend: Quantification of *Mdm2* in cholangiocytes (in the *Mdm2*-RNAscope images) shows the loss of *Mdm2* after the induction of the model. ** denotes $p < 0.01$, (Mean \pm SEM) Student's t-test, (N=3 per group)

In hepatocytes, the expression of *Mdm2* remains similar with and without induction at different time points (see below, **Supplementary Figure 2d**). Therefore, the presence of senescent hepatocytes is interpreted as the consequence of paracrine senescence, and not as primary senescence induced by Cre recombination or spontaneous loss of *Mdm2*.

Legend: Quantification of *Mdm2* in hepatocytes (in the *Mdm2*-RNAscope images) shows that hepatocytes maintain similar levels of *Mdm2* after the induction of the model. Student's t-test. (N=3 per group). The blue line depicts the level of *Mdm2* for both groups.

Furthermore, the levels of *Mdm2* in cholangiocytes after induction (see **Supplementary Figure 2b**) closely relate to the levels of recombination found in the K19Mdm2 model (assessed by the percentage of TdTom-

positive cholangiocytes, see **Supplementary Figure 7c**, depicted here below), suggesting that the cholangiocytes that have lost *Mdm2* are a product of recombination after Cre-activation.

Legend: Primary senescence in cholangiocytes is similar in both models. Images show YFP-positive cholangiocytes (green) and tdTom-positive (red) K19-positive cholangiocytes (grey). Upper corner of the image shows a digital magnification of representative areas. Far right, quantification of the number of YFP per K19-positive cholangiocytes (OPN-Mdm2^{flox/flox}YFP model) vs tdTom per K19-positive cholangiocytes (K19-Mdm2^{flox/flox}tdTom^{LSL}). Images show uninduced (Cre+ -TM) *versus* induced (Cre+ +TM) mice at day 2 after last tamoxifen injection.

We think the reviewer may also be suggesting that our results could be explained by cholangiocytes deleting MDM2 without activating the TdTom tracer and then becoming hepatocytes that are senescent- we cannot plausibly see this as an explanation given that cholangiocytes do not become hepatocytes unless there is strong selection pressure from hepatocyte injury and impairment of hepatocyte replication in the context of highly proliferative cholangiocytes (Raven et al. Nature).

10. Fig. 3c and 3d. Was p27 detected in bile ducts? According to the conclusion by the authors, p53+ primary senescent cholangiocytes do not express p27, but only p53- secondary senescent cholangiocytes express p27.

Yes, p27 is expressed in bile ducts as well.

We have included PCR gene expression analysis of *cdkn1b* in isolated bile ducts at day 2 after tamoxifen induction (Observe *cdkn1b* expression in **Supplementary Figure 9b**).

Legend: qRT-PCR of cell-cycle related genes *cdkn1a* (p21), *cdkn1b* (p27) and *cdkn2a* (p16) show a significant increase in isolated bile ducts of Cre+ +TM (N=6) group, when compared with the Cre- +TM group (N=5) (day 2 after induction). ** denotes $p < 0.01$, *** denotes $p < 0.001$ (Mean \pm SEM). Mann-Whitney test.

As the SASP can also act in an autocrine manner to reinforce senescence in the cell-of-origin (in this case, the primary senescent cholangiocytes)¹⁻³, expression of p27 in cholangiocytes could be associated with TGF β autocrine signaling.

We have changed the text in the manuscript to make it clear. It now reads ‘

‘p27, a direct target of Transforming Growth Factor β (TGF β), is also expressed in cholangiocytes and hepatocytes in a similar pattern to p21 (Fig. 3c and 3d).’...

‘TGF β can induce senescence in a paracrine manner, via a mechanism that generates reactive oxygen species and DNA damage^{11,17}. TGF β blocks the progression of the cell cycle at the restriction point in late G1 phase by transcriptional activation of cyclin-dependent kinase inhibitors (p21, p27 and p15,) through the SMAD complex^{11,17,18}.’

1. Hoare M and Narita M. Transmitting senescence to the cell neighbourhood. Nature Cell Biology. 2013; 15, 887-889.
2. Coppé, JP, Desprez PY, Krtolica A, Campisi J. The senescence-associated secretory phenotype: the dark side of tumor suppression. 2010; 5:99-118.
3. Salama R, Sadaie M, Hoare M, Narita M. Cellular senescence and its effector programs. Genes Dev. 2014. 15; 28(2):99-114.

11. Fig. 3h

Activation of TGF-beta signaling in hepatocytes should be confirmed.

We have now included RT-PCR data for TGFb receptor I and II in hepatocytes in uninduced vs induced K19MDM2 model. Significant increase of both TGFbRI and TGFbRII indicate an active TGFb signaling response in hepatocytes (**Supplementary Figure 9d**).

Legend: qRT-PCR of *Tgfb1* and *Tgfb2* in isolated hepatocytes of Cre- +TM (N=5) and Cre+ +TM (N=6) mice at day 2 after induction are significantly increased. * denotes $p < 0.05$, ** denotes $p < 0.01$ (Mean \pm SEM). Mann-Whitney test.

We have also included in **Supplementary Figure 9e** in situ hybridization (ISH) for *Tgfb1*. In this images, *Tgfb1* is expressed in cholangiocytes as well as in some hepatocytes after the induction of the model (Cre+ +TM Day 2). ISH for *Tgfb1* reveals *Tgfb1* expression in macrophages as well. It would be interesting in the future to assess the role of macrophages in the transmission of senescence in biliary injury.

Legend: ISH by RNAscope for *Tgfb1* in Cre- +TM and Cre+ +TM at day 2 (N=3 per group). For the Cre- +TM, the selected areas show the absence of *Tgfb1* in cholangiocytes. For the Cre+ +TM, selected areas are magnifications of cholangiocytes (bottom left) and hepatocytes (bottom right) that express *Tgfb1*. Notice the presence of some *Tgfb1*-positive macrophages in the Cre+ +TM group. Right *PPIB* positive control (N=1). Scale bars = 50 μ m

12. Fig. 4. The mouse line shows spontaneous ductular reaction without DDC (Sup Fig. 2a) . Was DDC-induced ductular reaction enhanced or reduce?

We have now included in **Supplementary Figure 10f** a comparison between Day 90 after induction of the model (S Day 90), Oil+DDC and Tamoxifen+DDC in the model.

Assessment of K19 staining shows that DDC-induced ductular reaction is enhanced in comparison with the rest of the groups. This could reflect an inability to proliferate of the senescent cells in response to DDC diet at short term.

Increase of Ductular reaction at day 90 in the model could reflect the effects of paracrine senescence in the surrounding cholangiocytes at long term.

Legend: Comparison of the total percentage of K19-positive cells. From left to right: day 90 after induction of the K19-Mdm2^{flox/flox}tdTom^{LSL} model; C57BL6 DDC (1 week DDC in wild type C57BL6 mice), and S+DDC (induced K19-Mdm2^{flox/flox}tdTom^{LSL} maintained 1 week under DDC diet).

13. The statement (line 113) “We also found increased numbers of αSMA-positive cells surrounding the senescent 114 cholangiocytes (Fig. 2g and Supplementary Fig. 3)” is not appropriate, because the authors measured fluorescence intensity of αSMA surrounding the bile duct. For this claim, the cell number should be counted.

Thank you for pointing this out. We have now changed from “numbers” to “intensity”, and the legend in the figure 2g reads: “We found increased intensity of αSMA-positive cells surrounding the senescent cholangiocytes (Fig. 2g, h).

We have also included the percentage of αSma-positive cells in Figure 2h to give an idea of the increase of the response of αSma-positive cells in the whole tissue.

Legend: Increased total percentage of αSMA-positive cells in the Cre+ +TM group at day 21.

14. Fig. 6c and 6d. Specificity of YFP+ biliary cells on the effect of inhibition of TGF-beta signaling needs to be shown.

Quantification of senescence markers is performed exclusively on the YFP-positive biliary cells (with or without LY2157299 treatment). To clarify this point, we have changed the text to:

‘Quantifications in the YFP-biliary cells showed increased senescent markers and decrease BrdU incorporation, demonstrates that senescence can be transmitted in a paracrine manner *in vitro* (Fig. 6b).

In order to inhibit the transmission of paracrine senescence, we blocked TGFβ signaling pathway and assessed the same markers of senescence in the YFP-biliary cells. To that purpose, we used a small molecule that targets the TGFβ receptors ALK4, ALK5 and ALK7²³(LY2157299), effectively blocking TGFβ signaling pathway (Fig. 6c).

Addition of LY2157299 caused a significant reduction of paracrine senescence in the YFP-biliary cells (Fig. 6d and Supplementary Fig. 7a, b),’

15. Fig. 7f. Effect of TGF-beta inhibitor. The authors state that the number of primary senescent cholangiocytes (tdTom-positive) remain unaltered, proving that primary senescence is not affected by the administration of LY2157299 (page 25, line 376) . However, tdTom+ cells were created by Cre-mediated recombination but was independent on senescence induction.

We wanted to indicate that administration of LY2157299 only blocks paracrine senescence. It is not designed

as a strategy to eliminate primary senescent cells (TdTom-positive). Whether primary senescence is affected by this compound, could be interesting to the field to investigate in the future. To clarify this point in the text, we have now changed to: 'Addition of LY2157299 caused a significant reduction of paracrine senescence in the YFP-biliary cells (Fig. 6d and Supplementary Fig. 7a, b), without altering the number of primary senescent biliary cells (Supplementary Fig. 7c).

16. Page 11, line 182 [other members of the TGF-beta family, such as Tgfr1 and Tgfr2] Tgfr1 and Tgfr2 are receptors, but not TGF-beta family members.

Thank you! We have now changed to: "After senescence, isolated bile ducts also exhibit increased gene expression of *Tgfr1* ($p=0.0023$) and *Tgfr2* ($p=0.0047$)."

17. In the *in vitro* system, macrophages removed senescent cells, whereas *in vivo* results (Fig. S2a) showed that tdTomato+ cells were not reduced. This discrepancy needs explanation.

Currently there is a trend to decrease in the number of total TdTom-positive recombined senescent cells in the *in vivo* model (see **Supplementary Figure 3a**). This result suggest that the recruited macrophages may actively target and eliminate senescent cells.

Legend: Quantification of the total percentage of TdTom-positive cells increases after induction of the model at day 2 but shows no significant differences over the course of 90 days. ANOVA, Sidak's multiple comparisons test.

We do observe infiltration of macrophages at day 5 and accumulation of mononuclear infiltrates at day 21. The elimination of senescent cells by immune surveillance has been described as a fast process in tumoural context^{1,2}. However, in our model, this period might be longer (as the mechanisms quite significantly differ). Alternatively, macrophages might target and engulf the apoptotic cells present in the model and then target the senescent cholangiocytes.

Our *in vitro* system represents a conceptualization of this process, in which we established a direct co-culture of senescent cholangiocytes and macrophages. This could potentially explain the fast target and elimination of senescent cholangiocytes (72 hours).

1. Kang et al. Senescence surveillance of pre-malignant hepatocytes limits liver cancer development. *Nature*. (2011). 479(7374): 547-51
2. Xue et al. Senescence and tumour clearance is triggered by p53 restoration in murine liver carcinomas. *Nature* (2007). 445(7128): 656-60.

REVIEWER 4.

The authors generated a model of biliary senescence by conditionally knocking out Mdm2 in bile ducts. Senescent cholangiocytes induced profound alterations in the microenvironment with recruitment of myofibroblasts and macrophages causing collagen deposition, TGFbeta production and induction of senescence in surrounding cells. This new model supports the concept of paracrine transmission of senescence. The authors identified TGFbeta as a major mediator of this response.

The in vitro and in vivo experiments are very convincing and provide a rationale for targeting TGFbeta to treat biliary disease and accompanying liver damage.

Comments:

1. The TGFbeta type I receptor (ALK5) inhibitor used in the study (LY2157299) is not specific as it will also block signaling through ALK4/ALK7 - this is an important point to include.

Thank you for pointing this out. We have changed the phrase to: "we used a small molecule that targets the TGFβ receptors ALK4, ALK5 and ALK7²³ (LY2157299) to block TGFβ signaling pathway".

2. In fact, it would be great for the field if the TGFbeta ligand(s) contributing to the paracrine transmission of senescence could be identified. The authors show that TGFbeta1 mRNA levels are elevated in their model, but what about TGFb2 and TGFb3, which utilise the same signalling pathway? Or activin A and activin B, which utilise ALK4/ALK7 and have previously been implicated in cholangiocyte development and liver regeneration. Showing which of these ligands contributes to senescence would be a major step forward, as it would enable the utilisation of more specific inhibitors.

We do completely agree. When we initially tested the values of TGFbeta family members (such as Tgfb2 and Tgfb3) there was a non significant trend to increase ($p=0.3203$ for *Tgfb2* and $p=0.4156$ for *Tgfb3*).

We have now included these previous data to **Supplementary Figure 9c**. We have also performed a new PCR for Inhibin A and B, which were non-significant as well.

Legend: qRT- PCR of other members of the TGFb signaling pathway, such as *Tgfb2*, *Tgfb3*, *Inha* (Inhibin a) and *Inhb* (Inhibin b) show no significant differences in isolated bile ducts of Cre+ +TM (N=6) group, when compared with the Cre- +TM group (N=5) (day 2 after induction). Mann-Whitney test.

We have now also included in **Supplementary Figure 9b** data about the expression of cell cycle genes such as *cdkn1a* (p21), *cdkn1b* (p27) and *cdkn2a* (p16) as positive controls to show that cell cycle stop is, in fact, established in isolated bile ducts in induced mice (Cre+ +TM) vs non-induced (Cre- +TM) at day 2 post-tamoxifen administration. We hope this will help to reinforce the idea that our model presents senescent cholangiocytes after tamoxifen induction.

Legend: qRT-PCR of cell-cycle related genes *cdkn1a* (p21), *cdkn1b* (p27) and *cdkn2a* (p16) show a significant increase in isolated bile ducts of Cre+ +TM (N=6) group, when compared with the Cre- +TM group (N=5) (day 2 after induction). ** denotes $p < 0.01$, *** denotes $p < 0.001$ (Mean \pm SEM). Mann-Whitney test.

We have also included in the **Supplementary Figure 9a** PCR controls for the isolation of bile ducts and hepatocytes in Cre- +TM mice at day 2 after tamoxifen induction. Isolated bile ducts expressed Keratin 19 (K19) but low levels of HNF4a. Conversely, isolated hepatocytes express HNF4a but no K19. We hope these data will give an idea about the purity of the populations isolated (as explained in the Methods section).

Legend: qRT-PCR analysis of *HNF4a* and *K19* in the isolated bile ducts (N=5) and hepatocytes (N=5) of Cre- +TM and Cre+ +TM mice day 2 after induction. Hepatocytes have high expression of the hepatocyte marker *HNF4a* but low expression of the epithelial marker *K19*. On the other hand, Bile ducts have low expression of *HNF4a* and high expression of *K19*. * denotes $p < 0.05$, ** denotes $p < 0.01$ (Mean \pm SEM). Mann-Whitney test.

3. If other TGFbeta ligands are upregulated, then their importance could be examined in the in vitro model. Please, refer to previous answer.

4. In Fig. 7B, it is difficult to see the p21+ cells - can these be expanded?

We have now taken new representative images magnification. We have also included an isotype control which we completely forgot to add previously. Hopefully this will help with the identification of the p21+ cholangiocytes.

Legend: Representative images of p21 (red) in cholangiocytes (green) in mice treated with vehicle, 10 or 20 mg/kg of LY2157299. (c) Percentage of p21-positive cholangiocytes diminishes with the administration of LY2157299.

5. Fig 6A & C - Should read "Quantify" and "Quantification" (in the actual figure)

We have now changed from Cuantify and Cuantification to Quantify and Quantification- many thanks!

FINAL REVIEWERS' COMMENTS:

Reviewer #1 (Remarks to the Author):

The authors have addressed all of my concerns and have incorporated all of the data into the manuscript as suggested.

Reviewer #2 (Remarks to the Author):

The authors have carried out a most thorough response to the questions raised by each of the 4 referees

Referee 1: they have looked at senescence in other K19 tissues, though the micrograph states p21, the legend states p16?? They have also responded to the query regarding disease progression and looked at Ki67 and p21 in controls.

Thank you for pointing this out. This is indeed a p16 staining. We have changed the panel to p16.

Referee 2: Explained apparent inconsistencies between quantitative data and Figs due to random selection of Figs and high inter-animal variability.

Referee 3: Justification for focussing on TGF β is given, along with acknowledgement that other factors may be involved in the SASP. Showed that proliferation arrest was irreversible, at least for up to 90 days. TUNEL staining showed that apoptosis was probably a secondary consequence, but do acknowledge that apoptosis may amplify the effect of senescence. Also exclude possibility that WT-1 is involved. Isotype controls now included for DCR and H2a immunofluorescence. Also show relative expression of mdm2 in cholangiocytes and hepatocytes, showing mdm2 is floxed out of many cholangiocytes, while mdm2 expression in hepatocytes remains very high with/without induction by Tm, suggesting recombination does not happen in hepatocytes. There was no evidence that cholangiocyte to hepatocyte metaplasia occurred, moreover it was considered highly unlikely that cholangiocytes could delete mdm2 without activating the TdTom marker and then undergo metaplasia. The TGF β signalling pathway was also confirmed by expressions of TGF β 1 and TGFR1 and R2.

Referee 4: The limitations (lack of specificity) of the inhibitors was acknowledged along with data on expressions of other TGF family members, though none appeared to be involved in this model. Better p21 images have been provided

Reviewer #3 (Remarks to the Author):

The authors have addressed my comments adequately. I have no further comments.

Reviewer #4 (Remarks to the Author):

The authors have addressed all of my concerns. In addition, they appear to have gone to considerable effort to answer the questions posed by the other reviewers. I think the manuscript is better for the new additions/clarifications. I recommend the article be published in Nature Communications.